# Insulin signaling mediates neurodegeneration in glioma

Patricia Jarabo[1] , Carmen de Pablo[1] , Héctor Herranz[2] , Francisco Antonio Martín[1],* , Sergio Casas-Tintó[1],*

Cell to cell communication facilitates tissue development and physiology. Under pathological conditions, brain tumors disrupt glia-neuron communication signals that in consequence, promote tumor expansion at the expense of surrounding healthy tissue. The glioblastoma is one of the most aggressive and frequent primary brain tumors. This type of glioma expands and infiltrates into the brain, causing neuronal degeneration and neurological decay, among other symptoms. Here, we describe in a *Drosophila* model how glioblastoma cells produce ImpL2, an antagonist of the insulin pathway, which targets neighboring neurons and causes mitochondrial disruption as well as synapse loss, both early symptoms of neurodegeneration. Furthermore, glioblastoma progression requires insulin pathway attenuation in neurons. Restoration of neuronal insulin activity is sufficient to rescue synapse loss and to delay the premature death caused by glioma. Therefore, signals from glioblastoma to neuron emerge as a potential field of study to prevent neurodegeneration and to develop anti-tumoral strategies.

## Significance Statement

Glioblastoma (GB) is among the most aggressive type of brain tumors, and currently there is no cure or effective treatment. Patients course with neurological decay and previous data in animal models indicate that GB causes a neurodegenerative process. In a *Drosophila* model, we describe here a molecule named ImpL2 that is produced by GB cells and impacts on neighboring neurons. ImpL2 is an antagonist of the insulin pathway, and insulin signaling reduction in neurons causes mitochondrial defects and synapse loss. These mechanisms underlying GB-induced neurodegeneration play a central role in the premature death caused by this tumor. Restoration of insulin signaling in neurons prevents tumor progression and rescues the lethality caused by GB in *Drosophila* models.

## Introduction

Cancer is one of the leading causes of mortality worldwide and is expected to be responsible for 15 million deaths in 2020 (65% in less

developed countries) according to the World Health Organization. Notwithstanding recent advances in health treatments and extended lifespan of patients, some tumors still remain incurable. Among them, GB stands out because it is the most frequent and a very aggressive primary brain tumor. It is originated from glial cells and causes death within the first year after diagnosis (Bi & Beroukhim, 2014), despite standard treatments such as resection, radiotherapy, and chemotherapy. This is accompanied by broad neurological dysfunctions (Messaoudi et al, 2015). Brain tumors cause cognitive decline and neuronal dysfunction (reviewed in Gehrke et al [2013] and Bergo et al [2016]). These cognitive defects are consistent with typical neurodegenerative-associated symptoms such as synapse loss and mitochondrial alterations (Granholm et al, 2010; Levenson et al, 2014).

*Drosophila melanogaster*, the fruit fly, has emerged as a reliable animal system to mimic human diseases such as cancer (Sonoshita & Cagan, 2017). The aim is to model cellular and molecular mechanisms of human diseases, to identify targets for eventual diagnosis and treatments of patients. The power of *Drosophila* genetics allows genetic and pharmacological screens that may be translated to medicine, particularly for neurodegenerative disorders (Casas-Tinto et al, 2011; Dar et al, 2012; Burke et al, 2017; Held et al, 2019; Portela et al, 2019a). In fact, a *Drosophila* GB model that recapitulates most of the human disease features has been developed and validated (Read et al, 2009, 2013; Read, 2011; Portela et al, 2019a, 2019b). This model is based on two of the most frequent mutations in patients, a constitutively active forms of the epidermal growth factor receptor (EGFR) and the phosphatidylinositol-3 kinase (PI3K) catalytic subunit p110α (PI3K92E) driven by the glial specific *repo-Gal4* (Read et al, 2009). This animal model has brought novel understandings into GB molecular mechanisms (Read, 2011; Read et al, 2013; Weng et al, 2017; Portela et al, 2019a, 2019b).

*miRNAs* are short noncoding *RNAs* that control gene activity mainly through post-transcriptional mechanisms. Recently, they have been linked to almost all biological processes and diseases, particularly cancer (Peng & Croce, 2016; Sander & Herranz, 2019). Concerning glioma, the *miR-200* family (which includes *miR-200*, *miR-141*, and *miR-429*) plays central roles in GB development, metastasis, therapeutic response, and prognosis (reviewed in Peng et al [2018]). Low levels of miR-200 are indicative of poor prognosis in GB (Men et al, 2014). In colorectal cancer and GB, low expression

---

[1]Instituto Cajal, Consejo Superior de Investigaciones Cientificas (CSIC), Madrid, Spain  [2]University of Copenhagen, Copenhagen, Denmark

Correspondence: scasas@cajal.csic.es; famartin@cajal.csic.es
*Francisco Antonio Martín and Sergio Casas-Tintó contributed equally to this work

levels of *miRNAs* correlates with up-regulation of *insulin-like growth binding protein 7* (*IGFBP7*) (Jones et al, 2015). Likewise, in GB there is a transforming growth factor beta-2 (TGFB2)–dependent increase in IGFBP7 protein levels (Pen et al, 2008). However, the mechanisms involved in IGFBP7 influence on GB progression and its regulation by *miR-200* remains unsolved. In *Drosophila*, *miR-200* and *IGFBP7* are represented by *miR-8* and *Imaginal morphogenesis protein-late 2* (*ImpL2*), respectively (Honegger et al, 2008). In juvenile stages, *miR-8* has been found to regulate glial cell growth and to promote synaptic growth at the neuromuscular junction (Morante et al, 2013; Loya et al, 2014).

In contrast, *Drosophila* ImpL2 induces cachexia, a systemic effect characterized by anorexia and metabolic alterations induced by other malignant tumors (Petruzzelli & Wagner, 2016). Secreted ImpL2 from epithelial tumor cells induces systemic organ wasting and insulin resistance by antagonizing insulin signaling (Figueroa-Clarevega & Bilder, 2015; Kwon et al, 2015). Interestingly, PI3K and *Drosophila* Ras homolog enriched in brain (dRheb), two members of the insulin pathway, induce the formation of synapses between neurons (a process known as synaptogenesis) in the *Drosophila* larval brain (Martín-Peña et al, 2006). Actually, it has been shown that AKT, also involved in insulin signaling, acts as a pro-synaptogenic element (Jordán-Álvarez et al, 2017). These data strongly support a role for insulin signaling in the regulation of neuronal synaptic activity in *Drosophila*. In mammals, a similar effect of insulin signaling on synaptic plasticity has been described (Knafo & Esteban, 2017). Notably, synapse loss is an early step in neurodegeneration (Sephton & Yu, 2015; Henstridge et al, 2016). We have recently re-evaluated GB as a neurodegenerative disease, showing that GB reduces the number of synapsis through wingless/frizzled 1 (wg/fz1) signaling (Portela et al, 2019b), equivalent to mammalian WNT pathway (Arnés & Casas Tintó, 2017). However, whether tumoral glial cells are able to modify insulin signaling in contiguous neurons, and consequently alter the number of synapses, is yet unknown.

To study the mechanisms of communication among malignant glial cells and neurons, we used a previously well-characterized Drosophila GB model that reproduces the oncogenic transformation of glial cells and lethal glial neoplasia in post-embryonic larval (Read et al, 2009, 2013) or adult brains (Portela et al, 2019a), leading to lethal glial neoplasia. We previously reported a reduction in the number of synapses in the neuromuscular junction (NMJ) of adult flies caused by GB progression (Portela et al, 2019b). This neuro-degenerative process is consequence of genetic modifications caused in glial cells. This phenomenon suggests that signals originated in the glial tumor can impact on neighboring neurons. Synaptogenesis is tightly regulated by PI3K, a main player in insulin signaling pathway (Jordán-Álvarez et al, 2017). Moreover, GB progression correlates with high levels of secreted molecules that decrease insulin pathway activity, such as ImpL2 (IGFBP7 in humans) (Pen et al, 2008; Jones et al, 2015).

In this report, we showed that secreted ImpL2 from glial-derived tumoral cells antagonizes insulin signaling in neighboring neurons, inducing a reduction in synapse number and consequently promoting neurodegeneration. *ImpL2* expression in GB cells is regulated by *miR-8*, thus linking functionally miRNA pathway with insulin signaling in a GB model. We described the function of ImpL2

as a mediator in GB–neuron communication, responsible for the reduction in synapse number and neurological defects. Indeed, we propose the insulin pathway as a core signal in GB progression and neurological decay. Finally, we propose neurodegeneration as a relevant factor in the lethality induced by GB.

# Results

## ImpL2 mediates GB progression and the associated neurodegeneration

To determine *ImpL2* mRNA expression levels in GB we performed qRT-PCR experiments. *ImpL2* mRNA showed an increase in GB samples as compared with control brains (Fig 1A). To discriminate between *ImpL2* expression in neuronal or glial (GB) cells, we used a *MIMIC GFP* reporter that reproduced faithfully *ImpL2* expression (Nagarkar-Jaiswal et al, 2015). Consistently, GB cells showed higher reporter *GFP* levels than control glial cells, which are restored to control levels upon *ImpL2* knockdown (Fig 1B–D and G). Therefore, these results validate the *ImpL2* RNAi tool and indicate that *ImpL2* is up-regulated in GB cells.

In a previous work, we established that tumoral progression depended on the formation of a network of protrusions (i.e., an expansion of the membrane surface) named tumor microtubules (TMs), similarly to human GB (Osswald et al, 2015; Portela et al, 2019b). Besides, in mammals and flies, the TM network required the *GAP43* and *igloo* gene functions, respectively (Osswald et al, 2015; Portela et al, 2019b). We also showed that GB progression requires c-Jun N-terminal Kinase (JNK) pathway activity (Portela et al, 2019b). The *Drosophila* JNK homolog Basket (Bsk) plays a central role in JNK signaling in normal and tumoral conditions (Fahey-Lozano et al, 2019). To determine if *ImpL2* up-regulation in glial cells requires TMs network expansion, or JNK pathway activity in GB cells, we down-regulated *igloo* expression to prevent TMs formation, or overexpressed a dominant negative form of *bsk* (*BSK*$^{DN}$) to block JNK activity. *ImpL2* expression levels were reverted to normal levels in both cases, indicating that glial *ImpL2* expression was dependent on TMs and JNK pathway activity, but not transcriptionally regulated by activated EGFR or Dp110 (Fig 1E–G).

To study the neurodegeneration associated to GB progression, we quantified the number of synapses in the neuromuscular junction (NMJ) of the adult flies with GB. NMJ is a stereotyped structure that allows counting the number of synapses in the synaptic buttons of the motor neurons by using anti-bruchpilot, a specific antibody that recognizes synapses unambiguously (see the Materials and Methods section for details). To determine the contribution of glial ImpL2 to synapse loss, we knocked-down *ImpL2* in GB cells and counted the number of synapses in adult NMJs. The results showed that Impl2 reduction in GB cells counteracted the reduction in the number of synapses observed in GB samples (Fig 2A–C and E). Next, to determine if ImpL2 is sufficient to decrease the number of synapses or it requires further features of GB cells, we overexpressed *ImpL2* in wild-type (wt) glial cells and quantified the number of synapses. The results show a decreased number of synapses in NMJs (Fig 2D and E), consistent with our previous

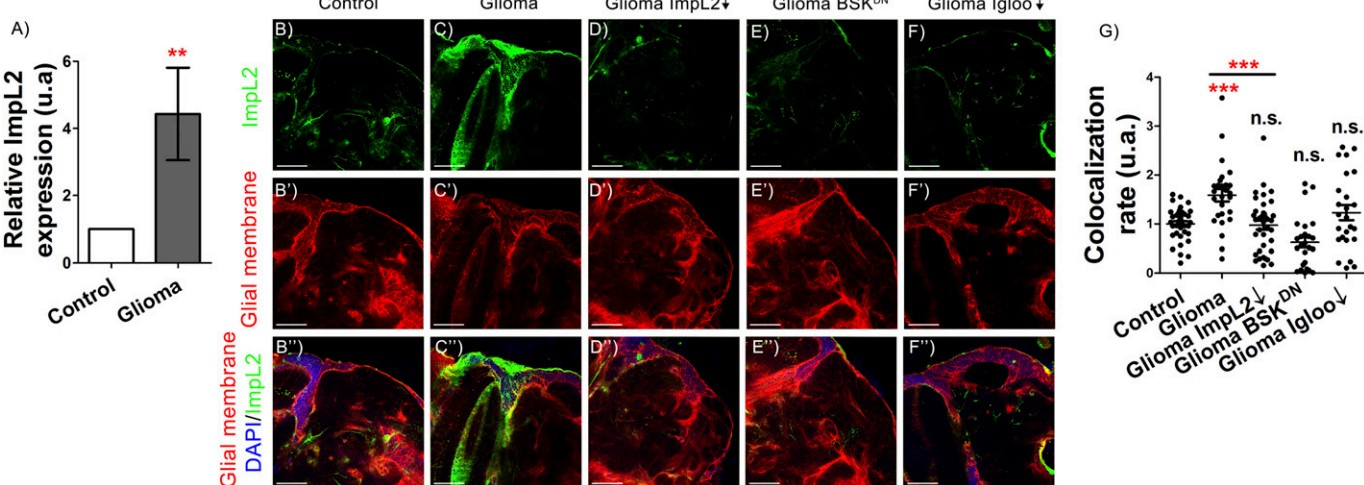

**Figure 1. *ImpL2* is up-regulated in glioma cells.**
**(A)** qRT-PCR of 7-d-old adult brain extracts from *repo>UAS-LacZ* (Control) and *repo>UAS-dEGFR*[λ], *UAS-dp110CAAX* (Glioma) flies shows an up-regulation of *ImpL2* in gliomas (*t* test). **(B, C, D, E, F)** Confocal microscopy images of 7-d-old adult brains from (B) *repo>UAS-LacZ* (Control), (C) *repo>UAS-dEGFR*[λ], *UAS-dp110CAAX* (Glioma), (D) *repo>UAS-dEGFR*[λ], *UAS-dp110CAAX, UAS-ImpL2 RNAi* (Glioma ImpL2↓), (E) *repo>UAS-dEGFR*[λ], *UAS-dp110CAAX, UAS-BSK*[DN] (Glioma BSK[DN]), and (F) *repo>UAS-dEGFR*[λ], *UAS-dp110CAAX, UAS-igloo RNAi* (Glioma Igloo↓) animals, in all cases combined with an *ImpL2-MIMIC* GFP transgene (scale bar, 50 µm). **(B', C', D', E', F')** Glial membrane is marked in red. **(B", C", D", E", F")** Merge and DAPI staining. **(G)** Quantification and statistical analysis of co-localization rate between ImpL2-MIMIC GFP and glial membrane in at least N = 10 per genotype (ANOVA, post hoc Bonferroni) (**P-value < 0.01, ***P-value < 0.001).

results, and suggest that *ImpL2* expressed in glial cells is sufficient to cause synapse loss in neurons.

Besides, we studied the two additional and typical features of GB such as the increase in the number of glial cells and the expansion of the TM network (Fig 2F, G, I, and J). The confocal images and quantifications showed that the down-regulation of *ImpL2* RNA levels in GB caused a striking reduction in the number of glial cells (Fig 2H and I) and in the total tumor volume (Fig 2J). Thus, we concluded that GB cells up-regulated and secreted ImpL2, a necessary step to induce tumoral expansion and a reduction of the synapse number in surrounding neurons. These results suggest that GB progression and neurodegeneration are coupled events.

### miRNAs levels inversely correlate with *ImpL2* expression and GB progression

It has been described that low levels of miRNAs correlated with high levels of ImpL2 homolog in human GB, suggesting that ImpL2 regulation might be mediated by miRNAs (Jones et al, 2015). Accordingly, there is a down-regulation of the *miRNA miR-200* family in GB samples (reviewed in Peng et al [2018]). Given that *miR-8*, the *Drosophila* homolog of *miR-200* family, negatively regulates *ImpL2* mRNA stability in the fat body (Lee et al, 2015), we hypothesized that *miR-8* might play a role in GB progression and ImpL2 levels regulation. We used a *miR-8* sensor to monitor *miR-8* activity in GB brains. It includes *miR-8* binding sites in the 3'UTR of the *GFP* gene (Kennell et al, 2012). Thus, high levels of GFP indicate low levels of *miR-8* activity and vice versa (see the Materials and Methods section). GFP signal was increased in GB cells (Fig 3A–C), indicating that *miR-8* levels were reduced. To determine if the increase in *ImpL2* in GB depended on *miR-8*, we analyzed *ImpL2* sensor upon *miR-8* overexpression in that context. We observed a significant

reduction in *ImpL2* expression in GB cells in vivo upon *miR-8* overexpression (Fig 3D–G). Consistently, *miR-8* gain-of-function in GB rescued the loss of synapses, recapitulating the effect of *ImpL2* loss of function in GB conditions (Fig 3H–J and L). These results are consistent with the effect of *ImpL2* down-regulation in GB that also rescued the synapse number (Fig 2C and E), and suggest that *miR-8* regulates *ImpL2* expression in GB.

Furthermore, *miR-8* overexpression in wt glial cells (with low *ImpL2* levels) did not alter the number of synapses (Fig 3K and L). Intriguingly, GB cell number expansion was not prevented by *miR-8* overexpression (Fig 3M–O and Q), and consistently *miR-8* overexpression increased glial cell number in wild type conditions (Fig 3P and Q). However, we did observe a reduction in GB membrane volume upon *miR-8* up-regulation, something that did not occur in normal glial cells (Fig 3R). Altogether, these results showed an inverse correlation between *miR-8* and *ImpL2* expression in GB cells and suggested that ImpL2 levels are regulated by *miR-8* in vivo.

### GB secreted ImpL2 reduces insulin signaling in neurons

To evaluate the impact of GB on insulin signaling in neurons, we used a fluorescent reporter (tGPH, composed by the fusion of a pleckstrin homology domain plus green fluorescent protein under the control of the tubulin promoter) that reports the activity of PI3K and, thus, is widely used as a monitor of insulin pathway (Britton et al, 2002). tGPH reporter activity in neurons is strongly reduced (although still detectable) upon GB induction compared with control neurons (Fig 4A and B). In addition, *ImpL2* knockdown specifically in GB cells prevents tGPH reporter reduction in neurons, suggesting a normalization of insulin signaling levels (Fig 4C).

To further analyze insulin-dependent FOXO activity, we used the *Thor*[MI09732] line that bears a *MIMIC* transgene inserted in the

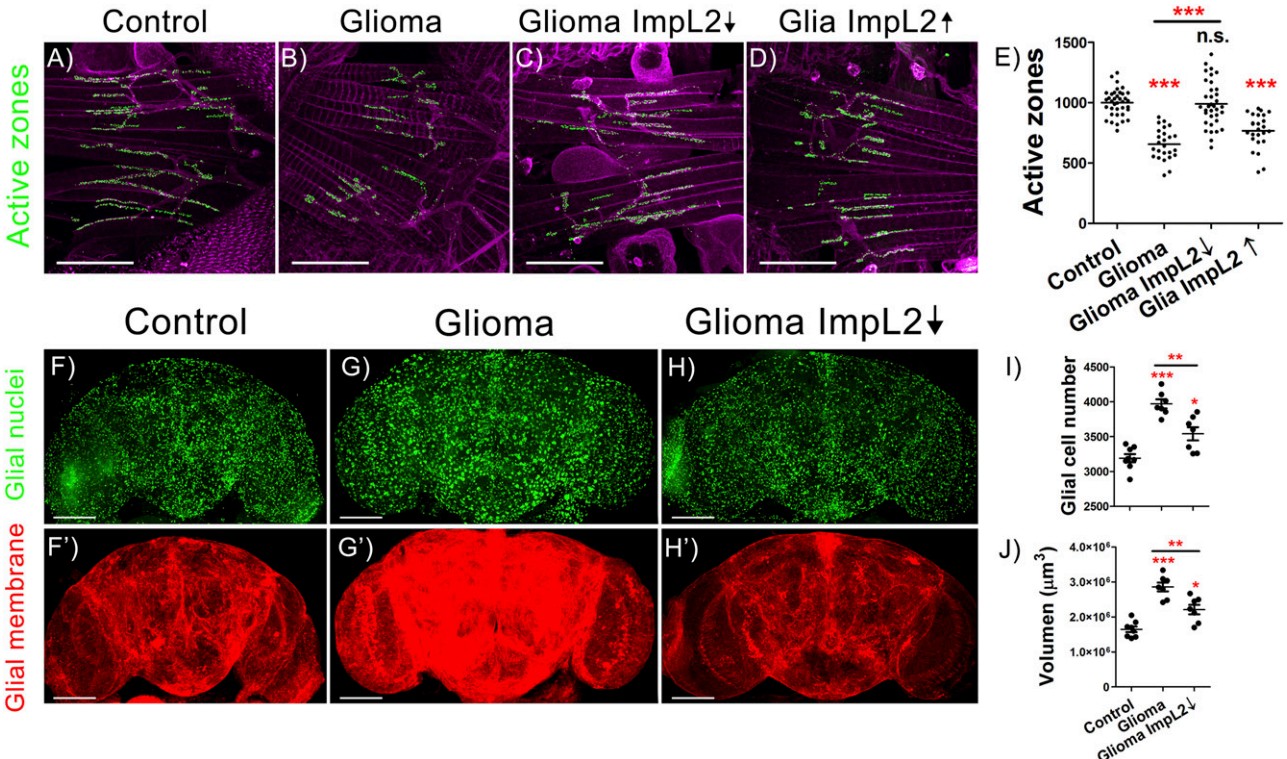

**Figure 2.** *ImpL2* **down-regulation in glioma cells rescues neurodegeneration and reduces tumor progression.**
**(A, B, C, D)** Confocal images of adult neuromuscular junction (NMJ) at 29°C from (A) *repo>UAS-LacZ* (Control), (B) *repo>UAS-dEGFR^A^, UAS-dp110CAAX* (Glioma), (C) *repo>UAS-dEGFR^A^, UAS-dp110CAAX, UAS-ImpL2 RNAi* (Glioma ImpL2↓), and (D) *repo>ImpL2* (Glia ImpL2↑) 7-d-old animals. Active zones are marked in green (anti-NC82) (scale bar, 50 µm). **(E)** Quantification and statistical analysis of active zones in at least *N* = 13 per genotype (ANOVA, post hoc Bonferroni). **(F, G, H)** Confocal microscopy images of adult brains from (F) *repo>UAS-LacZ*, (G) *repo>UAS-dEGFR^A^, UAS-dp110CAAX* and (H) *repo>UAS-dEGFR^A^, UAS-dp110CAAX, UAS-ImpL2 RNAi* flies after 7 d at 29°C with the glial nuclei marked in green. **(F', G', H')** Glial membrane is shown in red (scale bar, 100 µm). **(I, J)** Quantification of (I) glial cell number and (J) glial membrane volume for at least N = 7 per genotype (ANOVA, post hoc Bonferroni) (*P-value < 0.05, **P-value < 0.01, ***P-value < 0.001).

genomic region corresponding to the gene *Thor*. The *Thor* gene encodes for a protein that is involved in translational control. It is regulated by FOXO, and its expression can hence be used as a surrogate of FOXO activity. In normal conditions, *Thor* transcription remains at low but detectable levels (Teleman et al, 2005). However, when insulin activity is compromised, *Thor* is highly transcribed, as reflected by *LacZ* or *MIMIC* lines (Galagovsky et al, 2014). Neurons exposed to GB had reduced insulin signaling, as shown by *Thor^MI09732^* GFP expression. This effect on insulin pathway was restored by down-regulating *ImpL2* in GB cells (Fig 4D–G). All these results together suggested that *ImpL2* up-regulation in GB cells decreased the activity of insulin pathway in neurons, which is compatible with neurodegeneration.

However, the central function for insulin signaling pathway in synaptogenesis was described mainly in larval NMJ synapses (Martín-Peña et al, 2006). Whether or not insulin signaling plays a similar role in the central adult nervous system has not been evaluated yet. We expressed a dominant negative form of the *insulin receptor* (*InR^DN^*) in adult motor neurons and quantified the number of synapses. The quantification shows that *InR^DN^* expression in neurons caused a reduction in the number of synapses (Fig 4H–K). Moreover, the overexpression of *ImpL2* in glial cells also provoked a reduction in the synapse number. Thus, these results

together suggest that the ImpL2 effect on synapsis was due to a deregulation of the insulin signaling in neurons.

### Insulin signaling in neurons rescues partially neurodegeneration and mortality in GB

Our results suggest that in a GB, *ImpL2* overexpression causes an effective decrease in neuronal insulin pathway activity which in turn induces neurodegeneration. In consequence, we proposed that restoring insulin signaling specifically in neurons should prevent GB-induced neurodegeneration. However, to manipulate insulin pathway activity in neuronal population simultaneously with GB induction in glial cells, we needed to use the *LexA/lexAop* system (see the Materials and Methods section). We overexpressed *Rheb* (*lexAop-dRheb*), to activate insulin signaling in neurons under the control of an *elav-LexA* line. d*Rheb mRNA* levels increase significantly upon LexA/lexAop system activation, therefore validating the lexAop-d*Rheb* tool (Fig 5A).

The quantifications of adult NMJs showed an increase in the synapse number in NMJ when compared with GB and even to control genotypes, thus showing a similar effect (although slightly stronger) to GB with low ImpL2 levels (Fig 5B–D and E). In conclusion, the larval synaptogenic pathway regulated by insulin

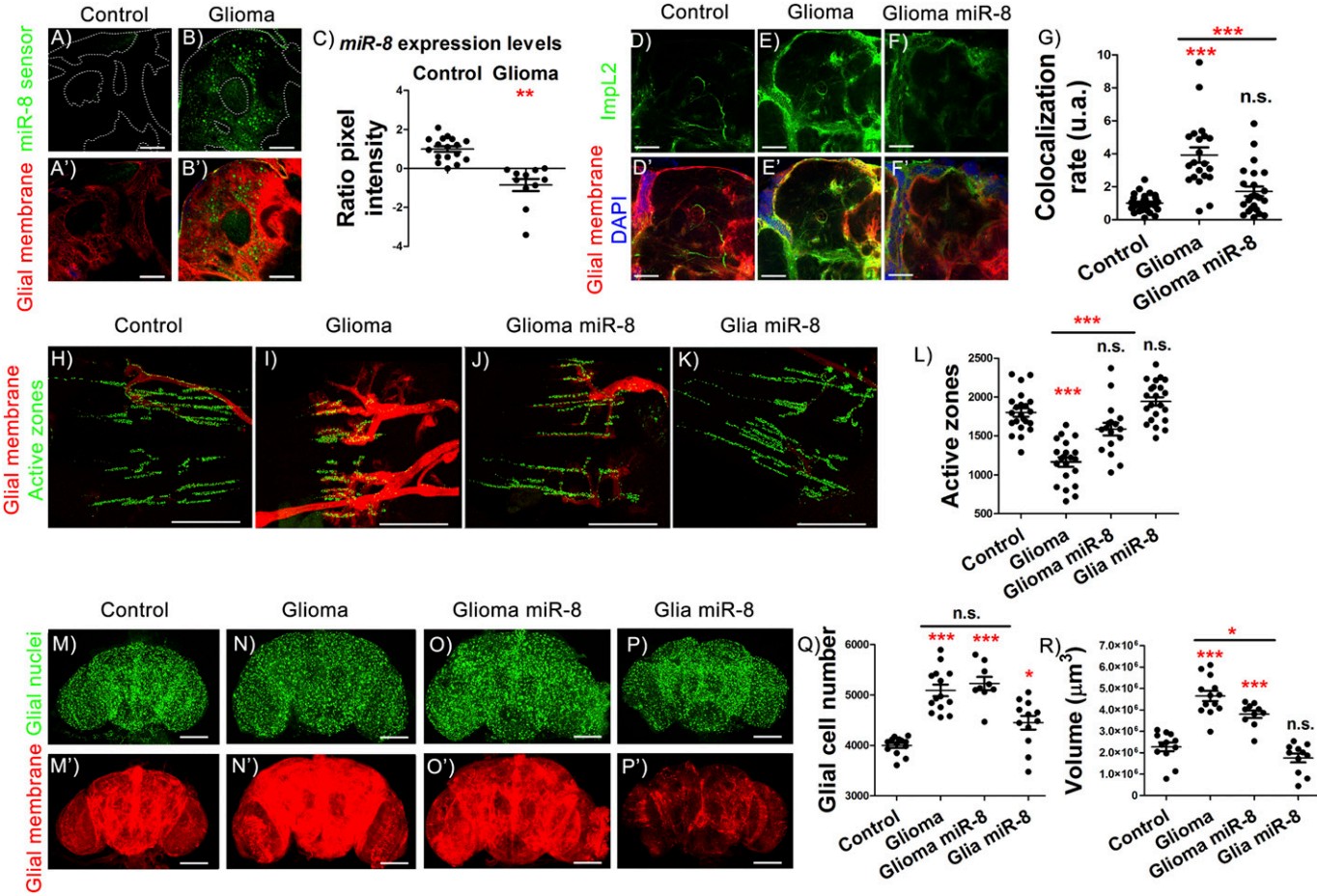

**Figure 3. Increasing *miR-8* levels in gliomas impairs *ImpL2* overexpression and synapse loss.**
**(A, B)** Confocal microscopy images of adult brain showing the expression pattern of *miR-8* in *repo>UAS-LacZ* control animals compared with *repo>UAS-dEGFR$^\lambda$*, *UAS-dp110CAAX* flies using *miR-8* sensor (see the Materials and Methods section). **(A', B')** Glial membrane is marked in red. **(C)** Quantification and statistical analysis of *miR-8* expression based on the GFP pixel intensity in at least *N* = 3 per genotype (ANOVA, post hoc Bonferroni) (scale bar, 50 *µm*). GFP control levels were normalized to one and GFP increase in glioma brains was transformed into negative values to indicate the relation between the increase in GFP and the reduction in *miR-8* levels of expression. **(D, E, F)** Confocal microscopy images of 7-d-old adult brains from the following genotypes: (D) *repo>UAS-LacZ* (Control), (E) *repo>UAS-dEGFR$^\lambda$, UAS-dp110CAAX* (Glioma), (F) *repo>UAS-dEGFR$^\lambda$, UAS-dp110CAAX, UAS-miR-8* (Glioma miR-8) animals, in all cases combined with ImpL2-MIMIC GFP transgene (scale bar, 50 *µm*). **(D', E', F')** Merge with glial membrane in red and DAPI staining. **(G)** Quantification and statistical analysis of co-localization rate between ImpL2-MIMIC GFP and glial membrane in at least N = 10 per genotype (ANOVA, post hoc Bonferroni). **(H, I, J, K)** Adult neuromuscular junction (NMJ) from (H) *repo>UAS-LacZ* (control), (I) *repo>UAS-dEGFR$^\lambda$, UAS-dp110CAAX* (Glioma), (J) *repo>UAS-dEGFR$^\lambda$, UAS-dp110CAAX, UAS-miR-8* (Glioma miR-8), and (K) *repo>miR-8* (Glia miR-8), active zones marked in green. **(L)** Quantification and statistical analysis of active zones in at least *N* = 10 per genotype (ANOVA, post hoc Bonferroni) (scale bar, 50 *µm*). **(M, N, O, P)** Confocal microscopy images of adult brains from 7-d-old flies of (M) *repo>UAS-LacZ*, (N) *repo>UAS-dEGFR$^\lambda$, UAS-dp110CAAX*, (O) *repo>UAS-dEGFR$^\lambda$, UAS-dp110CAAX, UAS-miR-8* and (P) *repo>UAS-miR-8* genotypes, glial nuclei marked in green. **(M', N', O', P')** Glial membrane is shown in red. **(Q, R)** Quantification of (Q) glial cells and (R) glial membrane volume for at least N = 9 per genotype (ANOVA, post hoc Bonferroni) (scale bar, 100 *µm*) (*P-value < 0.05**P-value < 0.01, ***P-value < 0.001).

signaling members was conserved in adult brains, at least for InR pathway and dRheb.

Insulin signaling mediates glucose metabolism and mitochondrial physiology (Cheng et al, 2010). In addition, mitochondrial alterations lead to synapse dysfunction and neurodegeneration (Chen et al, 2019; Zhao et al, 2019). In particular, the transport of mitochondria through the axons is altered in other neurodegenerative diseases (Vanhauwaert et al, 2019). Therefore, we investigated if GB progression affects neuronal mitochondria.

We used a *lexAop-mito-Cherry* reporter transgene to quantify the distribution of mitochondria in axon terminals. Pixel intensity quantification showed that GB causes a significant increase in mitochondria in the neuronal projections of Kenyon cells in the

mushroom body (Fig 5F–H, F'–H', and I), compatible with a neurodegenerative process (Chevalier-Larsen & Holzbaur, 2006). In line with this, we observed an increase in mitochondria accumulated in NMJ boutons that correlated with lower number of synaptic boutons (Fig 5F"–H" and J). Both effects in neuronal projections and NMJs were prevented upon insulin pathway signaling activation in the neuronal population of a GB brain (Fig 5H, H', and H"). Consistently, electron microscopy images of neuronal mitochondria in GB samples showed defects in the cristae, a typical feature of nonfunctional mitochondria (Long et al, 2015; Miyazono et al, 2018). This mitochondrial defective morphology was also reverted upon *dRheb* overexpression in neurons (Fig 5K–M). In addition, we quantified the size of the mitochondria in neurons and observed

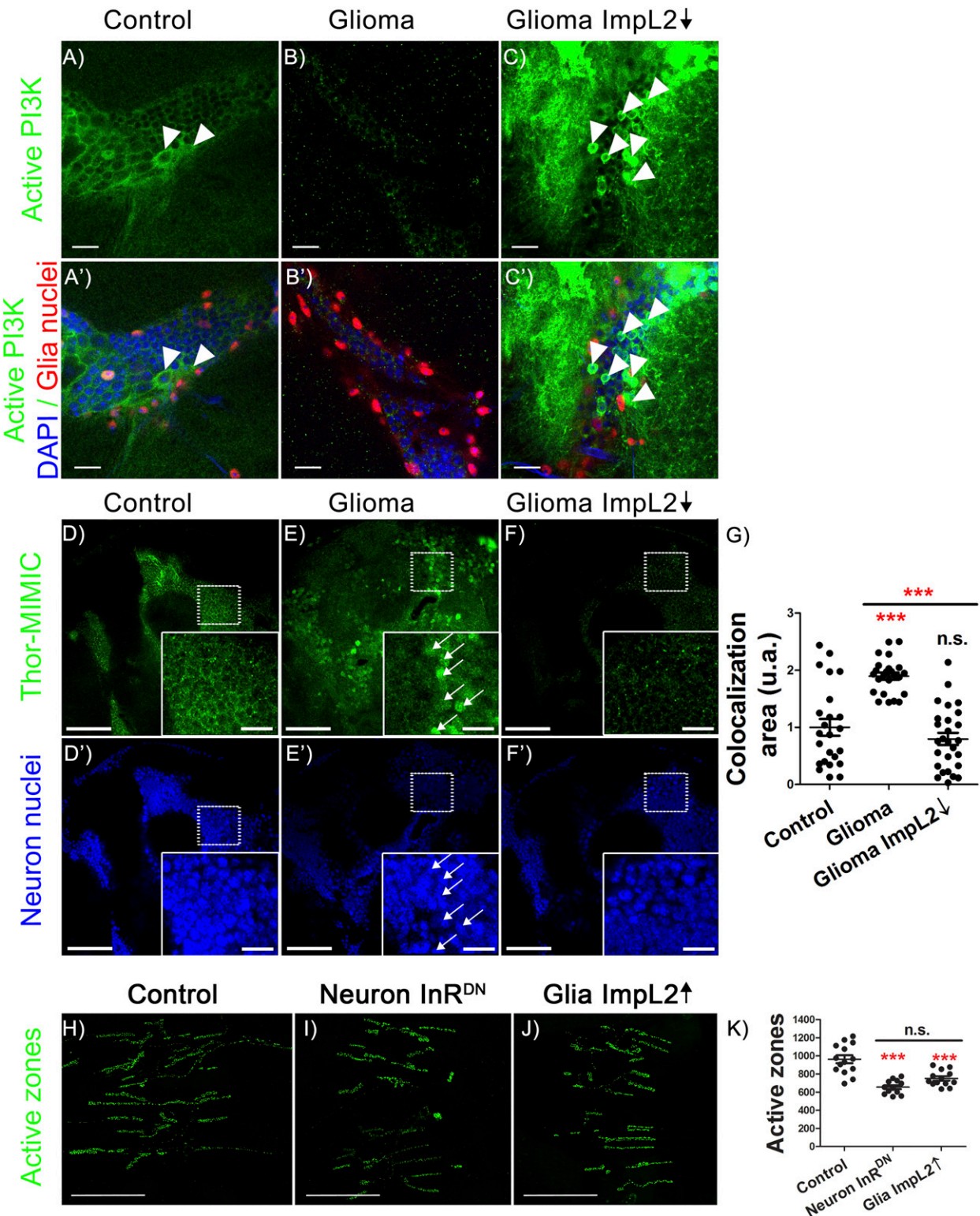

**Figure 4. Glioma-secreted ImpL2 inhibits insulin pathway activity in neurons.**
**(A, B, C)** Confocal microscopy images of adult brain of 7-d-old flies from (A) *repo>UAS-LacZ* (Control), (B) *repo>UAS-dEGFR$^\lambda$, UAS-dp110CAAX* (Glioma) and (C) *repo>UAS-dEGFR$^\lambda$, UAS-dp110CAAX, UAS-ImpL2 RNAi* (Glioma ImpL2↓) genotypes, combined with tGPH transgene reporter of PI3K activity (green). **(A', B', C')** Merge with glial nuclei in red and DAPI staining (scale bar 10 $\mu m$). White arrows point to non-glial cells with PI3K activity. The images are representative of at list N = 6 per genotype. **(D, E, F)** Confocal microscopy images of 7-d-old adult fly brains of the genotypes: (D) *repo>UAS-LacZ*, (E) *repo>UAS-dEGFR$^\lambda$, UAS-dp110CAAX* and (F) *repo>UAS-dEGFR$^\lambda$, UAS-dp110CAAX, UAS-ImpL2 RNAi*, in all the cases combined with *Thor$^{MI09732}$* line. **(D', E', F')** Neuronal nuclei are marked in blue. White arrows indicate nuclear localization of ThorGFP

that, in GB samples, mitochondria area is significantly reduced, and *dRheb* overexpression prevents this morphological defect (Fig 5N). Altogether, these results indicated that neurons showed mitochondrial disruption as a consequence of low insulin signaling. Increasing insulin pathway activity in neurons exposed to GB might be sufficient to recover functional organelles, as suggested by the synapse number.

GB brains with high insulin signaling in neurons also showed a reduction in the number of glial cells and in tumor volume (Fig 6A–E). More importantly, GB caused premature death, an effect that was significantly rescued by overexpressing *dRheb* in neurons (Fig 6F). These results suggested that restoration of neuronal insulin pathway activity improved the lifespan in animals with GB, thus linking synaptogenesis to a slower disease progression and functional protection. In conclusion, the communication between GB cells and neurons is proposed as a novel field of study for GB progression and provides a possible novel therapeutic target.

## Discussion

GB is one of the most aggressive type of brain tumor (Bi & Beroukhim, 2014). During GB progression, tumoral cells extend a network of membrane projections that contribute to brain infiltration and results in poor prognosis for the patient (Osswald et al, 2015). GB courses with a neurological decay that includes sleep disturbances, speech difficulties and other typical symptoms of neurodegeneration (Bergo et al, 2016). For decades, the origin of this decay was attributed to the high pressure caused by the GB solid mass and the associated edema. Despite the solid mass of the GB being removed after surgery, the neurodegenerative process continues, more likely because of the diffuse GB progression. This indicates that mechanisms underlying neurological decay are not restricted to the intracranial pressure and edema.

Recent publications suggested an active communication between GB cells and the surrounding healthy tissue, including neurons. Experiments performed with human GB cells in mice xenografts revealed a physical interaction between GB cells and neurons as electrical and chemical synapses (Venkataramani et al, 2019; Venkatesh et al, 2019). In this case, neurons act as presynaptic structures whereas GB tumoral cells are postsynaptic elements. These so-called "synapses" are required for GB progression. Besides, we have recently described cellular mechanisms for GB to deplete Wingless (Wg)/WNT from neurons. GB cells project TMs that enwrap neurons and accumulate Frizzled1 (Fz1) receptor to vampirize Wg from neighboring neurons. This unidirectional mechanism facilitates GB proliferation and causes a loss of synapses in the neurons (Portela et al, 2019b). In addition to this, here we also describe a bidirectional communication system between GB cells and neurons. In contrast to what happen with Wg/Fz1, ImpL2 protein is originated in GB cells and target healthy neurons, but not

vice versa. ImpL2 binds InR ligands and act as an antagonist of the pathway. In consequence, insulin signaling might be reduced in neurons which in turn caused synapse loss and lethality (Fig 7). Neuronal insulin signaling can be restored via *dRheb* up-regulation, and this is sufficient to extend the lifespan of GB animals. It is tempting to propose that ImpL2 impacts both tumor expansion (GB membrane) and neurodegeneration. The rescue of the neurodegenerative phenotype by *ImpL2* knockdown could be caused by an autonomous GB signaling that impairs tumoral progression, or by the restoration of insulin signaling in neurons. However, *dRheb* up-regulation in neurons prevented GB cells number increase, suggesting that GB progression is impaired when neuronal insulin signaling is active. These results suggest that reducing insulin signaling (and the subsequent neurodegeneration) is critical for GB progression and invasion, and ultimately for the lethality caused by GB. In conclusion, the GB is able to alter neuronal regular functions actively by at least two ways: vampirizing a required growth factor from neurons (like Wg) or secreting an antagonist of the pathway (such as Impl2). In both cases, the consequence is similar; a signaling pathway essential for synaptogenesis in the healthy neuronal population is down-regulated.

The regulation of *ImpL2* expression in GB cells seems to depend on miRNAs, at least in part. In cancer, miRNAs have emerged as general regulators key for tumoral progression, including GB (Sander & Herranz, 2019). Interestingly, *miR-200* family (*miR-8* in *Drosophila*) plays a key role in human GB. We have described a correlation between *miR-8 levels*, *ImpL2* expression and GB progression. However, there are no predicted *miR-8* binding sites in *ImpL2* sequence (STarMirSoftware for Statistical Folding of Nucleic Acids and Studies of Regulatory RNAs- (Rennie et al, 2014) or TargetSCAnFly- http://www.targetscan.org/); therefore, it is unlikely that *miR-8* regulates directly *ImpL2 mRNA* stability. One possibility is the existence of mediator proteins that depend directly on *miR-8*. The most evident possibility is a transcription factor whose mRNA stability is sensitive to *miR-8* and acts as a transcriptional regulator of *ImpL2*. However, the regulation of *ImpL2* and its association to miRNAs is a matter for future studies. Another intriguing point is the differential effects that both *miR-8* overexpression and *ImpL2* down-regulation have on GB growth. Whereas an excess of *miR-8*, which in turn reduces *ImpL2*, is unable to reduce GB cell number, direct down-regulation of *Impl2* significantly reduces the growth of GB cells. Nevertheless, it is known that most of miRNAs control several mRNAs, thus *miR-8* overexpression might affect to different extent other *mRNAs* than just *ImpL2* mRNA, which may account for such differences.

Altogether, our data suggest that the progression of brain tumors in *Drosophila* depends not only on the intrinsic properties of the tumoral cells, but also on the physiological condition of the surrounding cells (Fig 7). The potential relationship between Wg/WNT and insulin pathway has been proposed under physiological or tumoral conditions (Desbois-Mouthon et al, 2001; Yi et al, 2008;

---

signal. **(G)** Quantification and statistical analysis of co-localization area in *N* = 9 per genotype (ANOVA, post hoc Bonferroni) (scale bar, 50/10 μm). **(H, I, J, K)** Adult neuromuscular junction (NMJ) from the genotypes: (H) *D42>UAS-LacZ* (Control), (I) *D42>UAS-InR^DN* (Neuron InR^DN), and (J) *repo>UAS-ImpL2* (Glia ImpL2↑) animals after 7 d at 29°C (active zones shown in green). *D42* is expressed in motor neurons. **(K)** Quantification and statistical analysis of active zones in at least *N* = 6 per genotype (ANOVA, post hoc Bonferroni) (scale bar, 50 μm) (***P-value < 0.001).

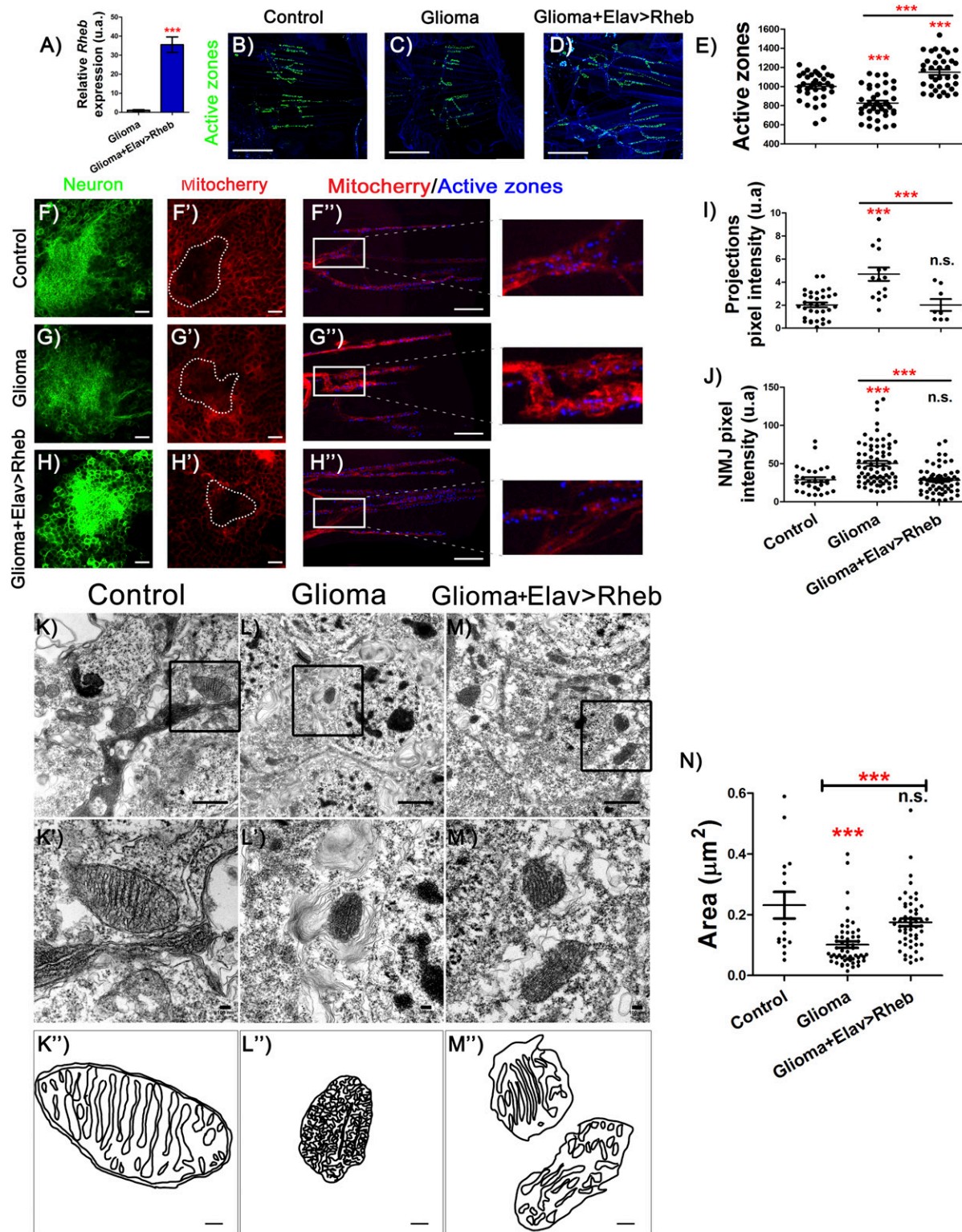

**Figure 5. Rheb expression in neurons rescues active zone loss and mitochondrial alterations.**
**(A)** qRT-PCR of 7-d-old adult brains from *repo>UAS-dEGFR^λ, UAS-dp110CAAX; Elav-LexA>lexAop-CD8GFP* (Glioma) and *repo>UAS-dEGFR^λ, UAS-dp110CAAX; elav-LexA>lexAop-CD8GFP, lexAop-Rheb* (Glioma+Elav>Rheb) flies show *Rheb* expression in glioma brains, and *Rheb* up-regulation by ectopic expression of *Rheb* in neurons (t test). **(B, C, D)** Adult neuromuscular junction (NMJ) from (B) *repo>UAS-LacZ; elav-LexA* (C) *repo>UAS-dEGFR^λ, UAS-dp110CAAX; elav-LexA* and (D) *repo>UAS-dEGFR^λ, UAS-dp110CAAX; elav-LexA lexAop-Rheb* after 7 d at 29°C (active zones are marked in green). **(E)** Quantification and statistical analysis of active zones in at least *N* = 20 per genotype (ANOVA, post hoc Bonferroni) (scale bar, 50 *μm*). **(F, F', G, G', H, H')** Confocal microscopy images of adult brains (detail of Kenyon cells) from (F) *repo>UAS-LacZ;*

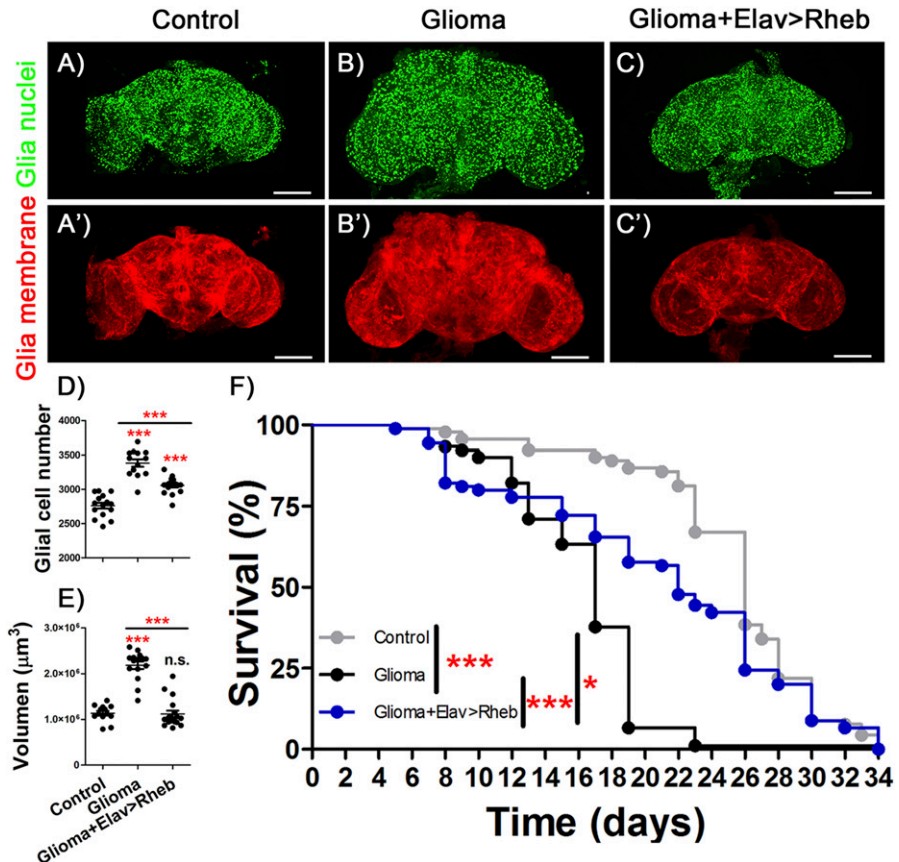

**Figure 6. Overexpression of *Rheb* in neurons protect against glioma effects.**
**(A, B, C)** Confocal microscopy images of adult brains from (A) *repo>UAS-LacZ*; *elav-LexA* (Control), (B) *repo>UAS-dEGFR^λ*, *UAS-dp110CAAX*; *elav-LexA* (Glioma) and (C) *repo>UAS-dEGFR^λ*, *UAS-dp110CAAX*; *elav-LexA, lexAop-Rheb* (Glioma+Elav>Rheb) after 7 d at 29°C with glial nuclei marked in green. **(A′, B′, C′)** Glial membrane is shown in red. **(D, E)** Glial cells number and (E) glial membrane volume quantification and statistical analysis for at least N = 13 per genotype (ANOVA, post hoc Bonferroni) (scale bar, 100 µm). **(F)** Graph shows a survival assay of *repo>UAS-LacZ*; *elav-LexA* (gray), *repo>UAS-dEGFR^λ*, *UAS-dp110CAAX*; *elav-LexA* (black) and *repo>UAS-dEGFR^λ*, *UAS-dp110CAAX*, *elav-LexA, lexAop-Rheb* (blue) male flies and statistical analysis in N = 90 (Mantel–Cox test) (*P-value < 0.05,**P-value < 0.01, ***P-value < 0.001).

Palsgaard et al, 2012) and represent a potential issue of interest to study in GB-host biology. We described recently (Portela et al, 2019b) the positive feedback loop established with wingless/JNK/MMPs and tumor microtubes that promote GB progression. We do not have evidences that *miR-8/ImpL2* regulation is directly controlled by EGFR and/or PI3K signaling pathways. However, the results included in Fig 1 suggest that a reduction in JNK pathway (BSK^DN) or the knockdown of *igloo* (prevention of TMs formation) reduces *ImpL2* expression. It has been described that *miR-8*-mutant animals activate JNK signaling, but there is no evidence that JNK pathway can regulate *miR-8*. However, miRNA regulation has the tendency to establish reciprocal feedback loops and networks (Herranz & Cohen, 2010), so it might be plausible that JNK signaling and *miR-8* would have such a reciprocal regulation. These results suggest that *ImpL2* up-regulation is sensitive to TMs formation and JNK, and one could speculate that it might be also dependent on Wg/WNT signaling pathway. Our observations in *Drosophila* suggest that both pathways (insulin and Wg) participate in the equilibrium between GB cells and neurons. The relations among all different pathways and the mutual regulation should be matter of study of future projects.

GB patients respond differently to the progression of the GB: some patients survive for a few months, whereas others survive for years. If we accept that the coordinated effect in GB and neurons result in differential tumor progression, the vast differences in how patients respond to GB could be, in part, dependent on genetic or epigenetic conditions related to InR signaling genes in neurons, and probably other pathways such as WNT or Hedgehog.

# Materials and Methods

### Fly stocks and genetics

All fly stocks were maintained at 25°C (unless otherwise specified) on a 12/12 h light/dark cycles at constant humidity in a standard

---

*elav-LexA* (G) *repo>UAS-dEGFR^λ*, *UAS-dp110CAAX*; *elav-LexA* and (H) *repo>UAS-dEGFR^λ*, *UAS-dp110CAAX*; *elav-LexA lexAop-Rheb* 7 d at 29°C in all the cases combined with *lexAop-mito-Cherry* transgene (red), and neural membrane shown in green and (F′, G′, H′), mitochondrial membrane in red (scale bar, 10 µm). **(F, F″, G, G″, H, H″)** NMJ confocal images of the same genotypes as in (F, G, H) show mitochondrial marked in red and active zones in blue (scale bar, 10 µm). **(I, J)** Quantification and statistical analysis of pixel intensity in (I) projections and (J) NMJ in at least N = 15 and N = 6, respectively per genotype (ANOVA, post hoc Bonferroni). **(K, K′, K″, L, L′, L″, M, M′, M″)** Electron microscopy images of neurons from adult brains of (K) *repo>UAS-LacZ*; *elav-LexA*, (L) *repo>UAS-dEGFR^λ*, *UAS-dp110CAAX*; *elav-LexA* and (M) *repo>UAS-dEGFR^λ*, *UAS-dp110CAAX*; *elav-LexA lexAop-Rheb* (scale bar, 1 µm K′, L′, M′), an amplification from black square of mitochondria in detail (scale bar, 100 nm) and cristae (K″, L″, M″) a schematic representation of the mitochondrial (scale bar, 100 nm). **(N)** Quantification and statistical analysis of mitochondrial area (***P-value < 0.001).

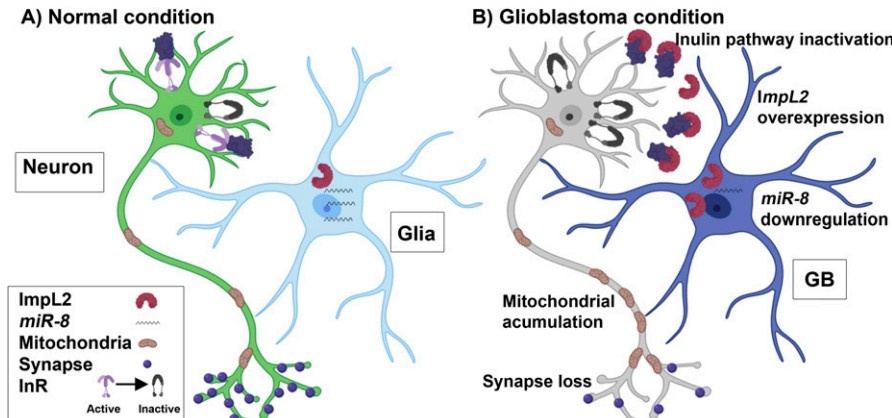

**Figure 7. Schematic representation of the effect of glioblastoma (GB) on neuronal insulin pathway.**
**(A)** In normal condition glial cells show physiological levels of *miR-8* and ImpL2. **(B)** In GB condition, *miR-8* expression is down-regulated, and *ImpL2* levels increase, which in turn causes an inactivation of the insulin pathway signaling in neurons. This effect triggers several changes in neurons such as synapse loss and mitochondrial accumulation throughout the axon. The three strategies addressed in this work to revert the tumoral phenotype consist in the up-regulation of *miR-8* expression in GB cells, the down-regulation of *ImpL2* also in GB cells and the activation of the insulin pathway in the surrounding neurons by means of *Rheb* overexpression.

medium. The stocks used from Bloomington Stock Center were *tub-Gal80$^{ts}$* (BL-7019), *Repo-Gal4* (BL-7415), *D42-Gal4* (BL-8816), *UAS-InR DN* (Bl-2852), *UAS-myr-RFP* (BL-7119) *UAS-LacZ* (BL-8529), *UAS-ImpL2 RNAi* (BL-55855), *ImpL2-MIMIC* (BL-59246), *lexAop-mito-Cherry* (BL-66530), tGPH (BL-8164), and *Thor$^{MI09732}$* (BL-53159). Other fly stocks used were *miR-8 sensor* (Kennell et al, 2012), *UAS-miR-8-RFP* (Lu et al, 2014), *Elav-LexA* (BL52676), *UAS-HRP::CD2* (gifted by L Luo), *UAS-dEGFR$^\lambda$;UAS-dp110$^{CAAX}$* (gift from R Read, Read et al, 2009) *UAS-ImpL2* (gift from Hugo Stocker), and *lexAop-Rheb* (gift from Nuria Romero).

The glioma-inducing line contains the *UAS-dEGFR$^\lambda$*, *UAS-dp110$^{CAAX}$* transgenes that encodes for the constitutively active forms of the human orthologs PI3K and EGFR, respectively (Read et al, 2009). *Repo-Gal4* line drives the *Gal4* expression to glial cells and precursors (Lee & Jones, 2005; Casas-Tintó et al, 2017) and combined with the *UAS-dEGFR$^\lambda$*, *UAS-dp110$^{CAAX}$* line allow us to generate a glioma thanks to the Gal4 system (Brand & Perrimon, 1993). *Elav-LexA* line drives the expression to neurons, allowing us to manipulate neurons in a glioma combining *LexA* and *Gal4* expression systems (Lai & Lee, 2006).

Gal80$^{TS}$ is a repressor of the Gal4 activity at 18°C, although at 29°C is inactivated (McGuire et al, 2003). The *tub-Gal80$^{ts}$* construct was used in all the crosses to avoid the lethality caused by the glioma development during the larval stage. The crosses were kept at 17°C until the adult flies emerged. To inactivate the Gal80$^{ts}$ protein and activate the Gal4/UAS system to allow the expression of our genes of interest, the adult flies were maintained at 29°C for 7 d except in the survival assay (flies were at 29°C until death).

Because of the inherent variability of tumor growth and the use of different reporters, we used the appropriate control and glioma genotypes that include them and performed the experiment in parallel for each grouped panel (at least three times): Figs 1B–F, 2, 3A, B, D–F, and –P, 4A–C and F–H, 5A–E and J–L, and 6.

### Immunostaining and image acquisition

Adult brains were dissected and fixed with 4% formaldehyde in phosphate-buffered saline for 20 min whereas adult NMJ were fixed 10 min; in both cases, samples were washed 3 × 15 min with PBS+0.4% triton, blocked for 1 h with PBS+0.4% triton+ BSA 5%, incubated overnight with primary antibodies, washed 3 × 15 min, incubated with secondary antibodies for 2 h, and mounted in Vectashield mounting medium, with DAPI in the case of the brains. The primary antibodies used were anti-repo mouse (1/200; DSHB) to recognize glial nuclei, anti-bruchpilot-NC82-mouse (1/50; DSHB) to recognize the presynaptic protein bruchpilot, anti-HRP rabbit (1/400; Cell Signalling) to recognize membranes, anti-GFP rabbit (1:500; DSHB) and anti-Elav (1:100; DSHB) to recognize neuron nuclei. The secondary antibodies used were Alexa 488 or 647 (1/500; Life Technologies). Images were taken by a Leica SP5 confocal microscopy.

### RNA extraction, reverse transcription and qRT-PCR

For RNA extraction, 1- to 4-d-old male adults were entrained to a 12:12 h LD cycle for 7 d at 29°C and then collected on dry ice at ZT 6. Total RNA was extracted from 30 heads of adult males of the Control (*repo>LacZ*), Glioma (*repo>UAS-dEGFR$^\lambda$, UAS-dp110$^{CAAX}$*), and *repo>UAS-dEGFR$^\lambda$, UAS-dp110$^{CAAX}$, elav-LexA, lexAop-Rheb* genotypes after 7 d of glioma development. RNA was extracted with TRIzol and phenol chloroform. Total RNA concentration was measured by using NanoDrop ND-1000. cDNA was synthetized from 1 mg of total RNA using M-MLV RT (Invitrogen). cDNA samples from 1:5 dilutions were used for real-time PCR reactions. Transcription levels were determined in a 14-ml volume in duplicate using SYBR Green (Applied Biosystem) and 7500 qPCR (Thermo Fisher Scientific). We analyzed transcription levels of *ImpL2*, *dRheb*, and *Rp49* as housekeeping gene reference.

Sequences of primers were RP49 F: GCATACAGGCCCAAGATCGT, Rp49 R: AACCGATGTTGGGCATCAGA, ImpL2 F: CCGAGATCACCTGGTTGAAT, ImpL2 R: AGGTATCGGCGGTATCCTTT, dRheb F:CGACGTAATGGG-CAAGAAAT, and dRheb R: CAAGACAACCGCTCTTCTCC.

After completing each real-time PCR run, outlier data were analyzed using 7500 software (Applied Biosystems). Ct values of duplicates from three biological samples were analyzed calculating 2DDCt and comparing the results using a *t* test with GraphPad (GraphPad Software).

### Viability and survival assays

Lifespan was determined under 12:12 h LD cycles at 29°C conditions. Three replicates of 30 1- to 4-d-old male adults were collected in vials containing standard *Drosophila* media and transferred every 2–3 d to fresh *Drosophila* media.

### Electron microscopy

Adult brains of *repo>LacZ*, *repo>UAS-dEGFR^λ*, *UAS-dp110^CAAX*, and *repo>UAS-dEGFR^λ*, *UAS-dp110^CAAX*, *elav-LexA>lexAop-Rheb* animals expressing CD2-HRP in glial membranes were dissected after 7 d of glioma development and fixed with 4% formaldehyde in phosphate-buffered saline for 30 min. The samples were washed twice with PBS and incubate with R.T.U. VECTASTAIN kit (VECTOR) for 30 min at RT and washed once with PBS. Followed by an incubation in dark with SIGMA FAST 3,3′-Diaminobenzidine Tablet SETS (Sigma-Aldrich) for 75 min at RT, washed once with PBT, and incubate with 4% formaldehyde + 2% glutaraldehyde for 1 h and stored at 4°C. Following fixation samples were washed three times in 0.1 M phosphate buffer. Post-fixation was performed in 1% osmium tetroxide + 1% potassium ferrocyanide for 1 h at 4°C, three washes in $H_2O_2dd$ and incubated in PBS 0.1M + 0.15% tanic acid for 1 min, washed once in PBS 0.1M, and twice in $H_2O_2dd$. After incubation in 2% uranil acetate, it was incubated for 1 h at RT in darkness and washesd three times in $H_2O_2dd$. Dehydration was done in ethanol series (30%, 50%, 70%, 90%, and 3 × 100%). The samples were infiltrated with increasing concentrations of epoxy resin TAAB-812 (TAAB Laboratories) in propilenoxid and encapsulated in BEEM capsules to polymerize 48 h at 60°C. Ultrathin sections of 70–80 nm were cut using Ultracut E microtome (Leica) and stained with 2% uranyl acetate solution in water and lead Reynols citrate. Grids were examined with JeolJEM1400Flash electron microscope at 80 kV. Images were taken with a OneView (4K × 4K) CMOS camera (Gatan).

### Quantification

Fluorescent reporter-relative ImpL2 and Thor signals within brains were determined from images taken at the same confocal settings avoiding saturation. For the analysis of co-localization rates, "co-localization" tool from LAS AF Lite software (Leica) was used taking the co-localization rate data for the statistics analyzing the co-localization between green signal (both cases) and signal coming from glial tissue (for ImpL2 levels) or neuronal nuclei (for Thor levels) from three slices per brain in similar positions of the z axis.

Average pixel intensity from *miR-8* sensor and *mito-Cherry* was measured using measurement log tool from Adobe Photoshop CS5.1. Average pixel intensity of *miR-8* sensor was analyzed quantifying the green sensor signaling glial tissue compared with green signal that did not overlap with glial cells to generate a ratio. Measurements were taken from similar localizations in three slices per brain with equivalent positions of the z axis. Average pixel intensity of mito-Cherry was analyzed taking measurements in the red signal from the reporter from at least five points of the NMJ.

Glial network was marked by a *UAS-myristoylated-RFP* reporter specifically expressed under the control of *repo-Gal4*. The total volume was quantified using Imaris surface tool (Imaris 6.3.1 software). Glial nuclei were marked by staining with the anti-Repo (DSHB). The number of Repo+ cells and number of synapses (anti-nc82; DSHB) were quantified by using the spots tool in Imaris 6.3.1 software. We selected a minimum size and threshold for the spot in the control samples of each experiment: 0.5 $\mu$m for active zones and 2 $\mu$m for glial cell nuclei. Then we applied the same conditions to the analysis the corresponding experimental sample.

For electron microscopy images quantification, we used FIJI (ImageJ 1.52v) software. After manually selecting the perimeter of each mito-chondrion, we measured the area, major and minor axis. All measurements were taken blind. In total, at least 15 mitochondria were measured from three different animals per genotype.

### Statistics

The results were analyzed using the GraphPad Prism 5 software (www.graphpad.com). Quantitative parameters were divided into parametric and nonparametric using the D'Agostino and Pearson omnibus normality test, and the variances were analyzed with F test. *t* test and ANOVA test with Bonferroni's post hoc were used in parametric parameters, using Welch's correction when necessary. To the nonparametric parameters, Mann–Whitney test and Kruskal–Wallis test with Dunn's post hoc were used. The survival assays were analyzed with Mantel–Cox test. The *P* limit value for rejecting the null hypothesis and considering the differences between cases as statistically significant was *P* < 0.05 (*). Other *P*-values are indicated as ** when *P* < 0.01 and *** when *P* < 0.001.

# Supplementary Information

# Acknowledgements

We thank Professor Alberto Ferrús for helpful discussions. We are grateful to R Read, the Bloomington *Drosophila* stock Centre and the Developmental Studies Hybridoma Bank for supplying fly stocks and antibodies, and FlyBase for its wealth of information. We acknowledge the support of the Confocal Microscopy unit and Molecular Biology unit at the Cajal Institute for their help with this project. We want to thank the continuous support from Emirates Khalifa Capital to this project. Research has been funded by grant BFU2015-65685P and PGC2018-094630-B-I00 (Ministerio de Innovacion y Ciencia [MICINN]). FA Martín is a recipient of a Ramon y Cajal Contract (RyC-2014–14961).

### Author Contributions

P Jarabo: formal analysis, funding acquisition, validation, investigation, methodology, and writing—original draft, review, and editing.
C de Pablo: formal analysis, investigation, and methodology.
H Herranz: formal analysis, supervision, funding acquisition, validation, investigation, visualization, methodology, and writing—original draft, review, and editing.
FA Martín: conceptualization, resources, formal analysis, supervision, funding acquisition, validation, investigation, methodology, project administration, and writing—original draft, review, and editing.
S Casas-Tintó: conceptualization, resources, formal analysis, supervision, funding acquisition, validation, investigation, visualization, methodology, project administration, and writing—original draft, review, and editing.

## Conflict of Interest Statement

The authors declare that they have no conflict of interest.

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
