## [Reviewer comments · Life Science Alliance]

Life Science Alliance

Insulin signaling mediates neurodegeneration in glioma

Patricia Jarabo, Carmen de Pablo, Héctor Herranz, Francisco Martin, and Sergio Casas-Tintó
DOI: <https://doi.org/10.26508/lsa.202000693>

Corresponding author(s): Sergio Casas-Tintó, Instituto Cajal and Francisco Martin, Instituto Cajal CSIC;

Review Timeline:

Submission Date:	2020-03-10
Editorial Decision:	2020-04-07
Appeal Received:	2020-08-21
Editorial Decision:	2020-09-08
Revision Received:	2020-12-01
Editorial Decision:	2021-01-04
Revision Received:	2021-01-07
Editorial Decision:	2021-01-08

Scientific Editor: Shachi Bhatt

Transaction Report:

April 7, 2020

Re: Life Science Alliance manuscript #LSA-2020-00693-T

Dr. Sergio Casas-Tintó
Instituto Cajal
Developmental Neurobiology
Avda Doctor Arce, 37
Madrid, Madrid 28002
Spain

Dear Dr. Casas-Tintó,

Thank you for submitting your manuscript entitled "Insulin signaling mediates neurodegeneration in glioma". The manuscript has been evaluated by expert reviewers, whose reports are appended below.

As you will see, the reviewers find your conclusions in principle interesting. However, they also raise many concerns, most of which pertain to data presentation and interpretation, as well as the quantifications performed. They further note overstatements and think that the data provided do not support all conclusions drawn. Given the reviewer input, we concluded that the robustness of the data is currently too unclear to move forward with your paper here. We have therefore decided to return your manuscript to you.

That said, the reviewers provide detailed and constructive input. Should you be able to address the concerns raised, we would be open to resubmission to Life Science Alliance of a significantly revised manuscript that addresses the concerns. Please note that we would need strong support on such a revised version from all reviewers.

Regardless of how you choose to proceed, we hope that the comments below will prove constructive as your work progresses.

Thank you for thinking of Life Science Alliance as an appropriate place to publish your work.

Sincerely,

Reviewer #1 (Comments to the Authors (Required)):

This is an interesting study of glia-neuron communication signals in the context of glioblastoma. The study reveals the role of ImpL2 in brain tumors and neurodegeneration, which potentially could be further studied to develop anti-tumoral strategies. An issue with the manuscript is that it is unclear whether miR-8 plays a major role in regulating ImpL2.

Specific comments:

1. The miR-8 part of the manuscript needs more supportive data. The authors claim that miR-8 may regulate ImpL2 indirectly in the discussion. However, to conclude that miR-8 plays a major role in ImpL2 regulation, the mechanism should be further explored to conclude that miR-8 regulates ImpL2 expression in GB.

2. In addition, there are a number of issues that need to be addressed:

- Figure 3G: I suppose 3G should be quantification of 3DEF and 3L should be quantification of HIJK. In the figure legend, 3G is a quantification of HIJK and 3L is quantification of MNOP which has no active zones.

- The correlation rate between ImpL2-MIMIC GFP and glial membrane in glioma is less than 2 in figure 1G but is 4 in figure 3G, and controls in these two experiments are same. Why the difference? If the correlation rate in glioma is 2, the miR-8 expression in glioma (figure 3G) has no significant effect. If the correlation rate in glioma is 4, the sample of figure 1G may have an issue. Or maybe this inconsistency is due to the quantification method?

Figures need to be improved:

3. Figure 1C': It is hard to tell where the glial membrane is due to strong GFP. Separate images would be better.

4. Figure 1G: If some ImpL2 staining does not co-localize with glia in gliomas, is ImpL2 also expressed in neurons?

5. Figure 1: What is the control of E and F? If B serves as control of C-F, the figures should be reorganized. Are these figures (B-F) made from same experiment?

6. Figure 2F-H: Is H a representative image? It seems that H has dramatically less glial membrane volume compared with control but in quantification (2J) it should be more? In addition, it is hard to compare the nuclei (green) due to their overlap with membrane (red). Separate images may provide more information, like in figure 3M.

7. All results should be presented in the past tense.

8. It would be better to use more informative word instead of e.g. "related". Sentences could be more concise and clearer. Here are few examples:

- "In juvenile stages, miR-8 has been related to glial cell growth and positively regulates positively synaptic growth at the neuromuscular junction (26, 27)." -- miR-8 has been found regulates glial cell growth and promotes synaptic growth at the neuromuscular junction...

- "In contrast, Drosophila ImpL2 is related to cachexia, a systemic effect characterized by anorexia and metabolic alterations induced by other malignant tumors (28)". --Drosophila ImpL2 induces cachexia...

- "However, the central function for insulin signaling pathway related to synaptogenesis was described mainly in larval NMJ synapsis (31)." Fix "related to" to "in".

- "Additionally, mitochondrial alterations are related to synapse dysfunction and neurodegeneration (48) (49)" Fix "are related" to "lead".

9. Figure 4: Insulin/TOR activity can be detected by an immunostaining of phosphorylated TOR target. This can provide more direct evidence of TOR activity and is a good addition to the transcriptional changes.

10. Figure 4E: Rp49 is not a proper housekeeping gene as activated insulin signaling/ overexpression of Rheb can lead upregulation of ribosomal proteins. Figure 4E may not reflect to

the actual levels of Rheb. Same issue in Figure 1A - although ImpL2 is likely upregulated.

11. Figure 5 J-L: Conclusion of these figures can only be made with a quantification of the phenotypes.

12. Figure 5: Insulin signaling has broad effects. Overexpression of Rheb completely reverted effects of glioma (5GHI), whereas the mitochondrial alterations seems to be only partially rescued. I wonder if some other functions of insulin signaling is critical for the phenotype of glioma but not the mitochondrial physiology.

Minor Issues (in order of appearance in the manuscript):

1. Line and page numbers would be helpful for the review process.

2. Abstract: "Therefore, signals from GB to neuron emerge..." should be fixed to "Therefore, signals from glioblastoma to neuron emerge..." or start using GB by the first "glioblastoma" word.

3. Introduction: "This model is based on two of the most frequent mutations in patients, a constitutively active form of the epidermal growth factor receptor (dEGFR λ) and the phosphatidylinositol-3 kinase (PI3K) catalytic subunit p110 α (PI3K92E) driven by the glial specific repo-Gal4 (16)", dEGFR λ should be EGFR.

4. "in GB development, metastasis, therapeutic response, and prognosis (reviewed by (21))." - "and prognosis (reviewed in 21)"

5. "We have recently re-evaluated GB as a neurodegenerative disease, showing that GB reduces the number of synapsis through wingless/frizzled 1 (wg/fz1) signaling (Portela et al, PLOS Biol 2019), equivalent to mammalian WNT pathway (36)". The reference should be fixed. Same issue in the first paragraph of results section.

6. "However, whether tumoral glial cells are able to modify insulin signaling directly in neurons, and consequently alter the number of synapses, is yet unknown." I suppose the "directly" in this sentence should be "remotely"?

7. In figure legends: "***p-value>0,005, ***p-value>0,0001" should be p-value<0,005 and p-value<0,0001. Same mistakes can be found in other figures.

8. "Consistently, GB cells show higher GFP levels than control glial cells. Likewise, upon ImpL2 RNAi expression we detect a decrease in GFP levels, similar to the ones observed in control brains (fig 1B-D)." This sentence needs rephrase.

9. Figure 1 G: It seems that the N is more than 10 in each genotype as shown by the number of dots in the figure. Should be fixed to the correct number. Same issue in Figure 2E, 3G, 3L, etc.

10. Figure 2: In figure title, "ImpL2 downregulation in glioma cells causes neurodegeneration and reduces tumor progression". Isn't that ImpL2 knockdown counteracted neurodegeneration?

11. "The GB itself is induced by overexpressing a constitutively active form of PI3K, thus the insulin pathway is activated in all glial cells. However, mRNA levels of dRheb are reduced in GB brains when compared with control brains, suggesting that this increase reflects mostly neuronal expression (fig 4E)." This conclusion is not clear to me, what is "this increase" in neuron?

12. Discussion: "GB is the most aggressive type of brain tumor." The word "most" is too strong, I suggest rewording to "one of the most".

Reviewer #2 (Comments to the Authors (Required)):

Insulin signaling mediates neurodegeneration in glioma

Patricia Jarabo, Carmen de Pablo, Héctor Herranz, Francisco Antonio Martín and Sergio Casas-Tintó1

In this interesting, and well-written, study, Jarabo and colleagues authors show that the secrete

Impl2 signal from glioblastoma-like cells dampens insulin signalling leading to neurodegeneration accompanied or caused by mitochondria alterations. The authors also show that overexpression of dRheb, a gene that was significantly downregulated in GB brain, specifically in neurons could rescue neuronal degeneration caused by glioblastoma cells and, more strikingly, suppressed tumorigenesis and rescued GB-mediated lethality.

Most of the conclusions are well supported and the figures are of quality. Moreover, if the mechanism is shown to be conserved in other animal species such as mice, the findings may open a new direction to study the glioblastoma-microenvironment interactions and the fly GB model could be of utility to future studies to integrate the numerous pathways affecting the in vivo invasion of GB cells.

I have some questions/comments or concerns to the authors:

(1) Page 6. Impl2 mediates GB progression and neurodegeneration

Figure 2 A-C, E. Here the authors cannot distinguish between the effects of Impl2 on tumorigenesis or those non-autonomous on neurodegeneration.? Impl2 downregulation reduced glioma cell membrane expansion, a feature of GB.

However, later, they show that in wt glial cells overexpression of Impl2 alters the number of synapses. While the data are a correlation, the authors could present the data indicating that together the most parsimonious explanation is that Impl2 impacts both tumorigenesis and neurodegeneration.

Note that this is different to the situation in the cancer-cachexia models

(2) Figure 2. "Downregulation of Impl2 causes neurodegeneration..."

Shouldn't it state that the opposite? Impl2-RNAi rescued the number of active zones and thus rescues neurodegeneration?

Indeed, in results the authors state " The results show that Impl2 reduction in GB cells counteracted the reduction in the number of synapses-synapses of GB brains"

This inconsistency needs to be corrected.

(3) MicroRNAs regulates... It should say miRNA regulated

(4) miR-8-Impl2. This needs further validation. Authors should more directly measure the levels of mir-8. While the ability of overexpression of mir-8, a known regulator of Impl2, has an impact, this alone is not sufficient evidence of the contribution of mir-8 in GB.

a. Analysis using either SP-mir-8, or epistasis using mir-8 mutations should be added to corroborate this conclusion. It would be expected that depletion of mir-8 would further increase Impl2 levels, connecting Impl2 to endogenous mir-8

(5) I have a question about the quantifications of miR-8 experiments. When comparing all graphs of Active zones, I noted that control in the miR-8 experiment show ~1500 active zones, whereas in the other graph is ~1000 active zone. This is a significant difference which questions whether the validity of the conclusions based on this graph? Could the authors elaborate on this discrepancy in the numbers in the various controls.

(6) Is the data representing the same genotypes in the different figures? If so, this should be explained

(7) GB secreted Impl2 reduces neuronal Insulin signaling

The authors state " To evaluate the impact of insulin signaling reduction in neurons, we measure dRheb mRNA by qPCR. dRheb is the molecular link between insulin signaling and TOR kinase, and

it reflects the insulin pathway activity (reviewed in 42)"

Nothing in this review supports this statement. Insulin and nutrient regulation of Rheb/mTORC1 signaling relies on activation of Rheb via subcellular location not transcription. What is the evidence that mRNA of Rheb reflects IIS activity? Saucedo (2003) has shown that mRNA Rheb is elevated in protein starved animals, but not in fed animals. This result does not support, by itself, that levels of mRNA Rheb is a proxy of IIS activity

(8) Page 9, the authors state " However, mRNA levels of dRheb are reduced in GB brains when compared with control brains, suggesting that this increase (?) reflects mostly neuronal expression (fig. 4E). What increase? If levels are decreased. What is the evidence that the change reflects mostly the neuronal expression?

If Rheb mRNA levels are inversely correlated with amino acid levels (Saucedo, 2003), it would be expected to be also inversely correlated to IIS activity. The observation that levels are reduced does not support the authors' claim of IIS reduction in GB brain neurons.

More, the authors must explain what evidence supports that the observed reduction of mRNA of dRheb in GB brains is brought about by ImpL2?

(9) Transcriptional regulation of THOR reflects dFOXO activity, not dTOR control because dTOR regulates THOR protein by phosphorylation and this is not examined here.

(10) The authors state: "neurons confronted with GB cells have reduced insulin signaling". Since the images in Figure 4 only shows the positive dots of THOR-MiMIC with respect to Elav, it seems appropriate to eliminate "confronted" of the text.

(11) 'All these results together suggest that ImpL2 up-regulation in GB cells mediates the decreased insulin pathway activity detected in neurons"

This is an unnecessary overstatement. The data are suggestive of a likely paper of IIS in neurons, and forcing conclusions by over-interpreting does not help. Moreover, the status of IIS in neurons need to be confirmed more convincingly.

(12) Images in Figure 5D-F and D'-F' have no resolution to see single mitochondria and to make any conclusion. Can the authors explain in what sense one would expect that the increased fluorescence intensity in the Cherry-mito GB brain to reflects neurodegeneration?

(13) Figure 5J-L We need here quantification, given that the analysis with fluorescence Cherry-mito did not yield sufficient resolution. The image of dRheb brain is unconvincing. The number of brains / cases of defective mitochondria and 'rescue' has not been included.

(14) Figure 6. Glioma elav>Rheb the size of this brain is almost as control. This is a rather intriguing observation which should be further supported. Ideally, IIS should be manipulated more directly via Pi3k/Akt/Pten in neurons and the status of pAkt and not mRNA of Rheb be assessed.

Discussion:

(15) Finally, here we also describe a one-way communication system from GB cells towards neurons.

This statement is probably incorrect because the manipulation of IIS via dRheb in neurons suppressed GB progression suggesting that communication is bi-directional as seen previously by others.

(16) ImpL2 binds DILPs and this may reduce insulin signaling in neurons. The rescue of 'mitochondrial aberration' is not convincing.

(17) We have described the presence and relevance of miR-8 in GB progression as a regulator of ImpL2 expression. This conclusion is based on correlative data and as such should be described in that way. Epistatic analysis could support and verify this idea.

(18) Authors should discuss in an inclusive way the potential relationship between neurodegeneration by WG/WNT and IIS

Minor:

- o 'the aim is not to heal a sick insect' I feel that this sentence is unnecessary. Those who might think this way are unlikely to be readers of this study
- o In Page 5. Portela et al 2019, eliminate PLOS Biol
- o Methods eliminate UAS- in ImpL2-MI14001
- o As far as I know all available mir-8-sensors are driven by the tubulin promoter not the UAS. And this include the sensor in reference 41. I could be wrong, but I suggest the authors to check this, too.
- o Fig. 5G. The order of this figure should be rearranged panel G is discussed before D
- o It should be corrected as Cherry-mito because in this construct is the N-terminus that is tagged with mCherry.
- o Please, add the citations to the statements on human GB in the first sentences of the Discussion
- o The titles shouldn't say something like: Reduced Insulin Signaling Mediates ...

Reviewer #3 (Comments to the Authors (Required)):

The authors propose here that glia overexpressing activated EGFR and PI3K secrete the insulin antagonist ImpL2, leading to reduced insulin signaling in neurons, and thereby phenotypes such as tumor progression and reduced neuronal synapses.

Overall, this is an interesting story. However, as detailed in 'major issues' below, several key findings are not solid, and some results don't seem to fit together. Furthermore, the manuscript is put together in a very sloppy way, with incorrect citations, lacking figure citations, and a lack of information on how assays were done in the figure legends and M&M, which makes it difficult to follow and to assess. These major issues should be addressed prior to publication.

Major Issues:

1. If activation of EGFR and PI3K in glia leads to cell-autonomous inhibition of mir8 expression and thereby induced ImpL2 levels, why do knockdown of basket or igloo in glia block this effect? Just because these two genes are required for formation of tumor microtubules, and thereby tumor progression, doesn't mean they should block the cell autonomous signaling pathway from EGFR+Dp110CAAX to mir8 to ImpL2? Is mir8 regulated cell autonomously by JNK signaling? If so, why? Do tumor microtubules regulate mir8 expression somehow? How?
2. According to the model, inhibition of mir8 by EGFR+Dp110CAAX leads to increased ImpL2 levels and thereby 3 phenotypes: reduced synapses, increased glial cell number, and increased glial cell membrane volume. Indeed, knockdown of ImpL2 rescues these three phenotypes (Fig 2). Mir8 overexpression seems to efficiently block the increase in ImpL2 levels in the tumors (Fig 3G). Then why doesn't it revert all the phenotypes - ie not glial cell number? The authors propose that mir8 overexpression also regulates some other gene that promotes glial proliferation. That's possible, but surprising. Unfortunately, from the materials & methods it is not clear how exactly glial cell number was measured/quantified. Were the tissues stained with a nuclear marker, which was then segmented and quantified, or was the glial membrane marker used to somehow count cell number? (The latter seems to be the case from the description). If so, how can changes in cell number be distinguished from glia that just have more or less membrane? In sum, it is not clear whether glial cell number is quantified correctly, and hence whether these conclusions are solid.
3. It is not clear whether the mir8 overexpression experiments (Fig 3) are overexpressing mir8 to physiological levels (ie restoring it to wildtype glia levels) or to very high supraphysiological levels.

This could be quantified, to show that the results are physiologically relevant. Alternatively, since mir8 null mutants are viable, the contribution of mir8 can be rigorously tested by measuring the phenotypic readouts (eg active zones) in a mir8 knockout background (ie compare mir8KO versus mir8, repo>EGFR+dp110CAAX)

4. The drop in insulin signaling levels in neurons is not solid:

-The authors write "dRheb mRNA levels drop down when Insulin signaling is low (43)." but I did not find this in the cited reference.

-The Rheb Q-RT-PCR result (Fig 4E) is strange. Presumably, according to the authors' interpretation, it should be increasing in the glia where Dp110 is activated, and decreasing in the neurons as a result of the secreted ImpL2 which reduces insulin signaling. However, overall, Rheb mRNA levels seem to be dropping to less than 5%. How can that be?

Here is one example (besides the ones listed below) of a poorly described experiment. In the figure legend, it says "RT-qPCR of Rheb expression is downregulated in gliomas". Is only the glioma being dissected out for the assay? I assume not (it would probably require laser capture), and that RNA is being extracted from the entire brain? In which case these are Rheb levels in the brain, not in the glioma.

In that case, is only 5% of the brain composed of glia? This seems unlikely. Even in that case, it would require Rheb mRNA levels to drop essentially to zero in the neurons.

How do the authors know that the Rheb they are measuring is neuronal Rheb and not glial Rheb?

-Fig 4A-C: it is not clear what is being imaged/analyzed here. Are these neurons or glia? According to the author's interpretation, insulin signaling should be high in glia (due to repo>Dp110CAAX) and low in neurons. Hence it makes a difference if Thor levels are being observed in neurons or glia. However there are no markers for neurons or glia in the image, and it is not even clear which region of the brain is being analyzed.

-One way to solidify this would be to look at a GFP-based PtdIns(3,4,5)P3 sensor, which has cellular resolution, and hence can be combined with glial/neuronal markers.

Minor Issues:

1. The abstract is misleading as currently written. The authors write "Here we describe how the glioblastoma produce ImpL2, an antagonist of the insulin pathway, which is regulated by the microRNA miR-8."

It should be made clear in the abstract that what is being studied is a drosophila model for glioblastoma, and not actual glioblastoma.

2. The first paragraph of the Results section is actually still introduction.

3. Legend to Figure 1:

-Legend to 1A should indicate whether the glioma is being dissected out (as suggested by the legend "upregulation of ImpL2 in gliomas") or whether this Q-RT-PCR is being done on whole brains or whole heads?

- "B)" is missing at the beginning of the description of panel B.

4. In Figure 1C-C' I cannot see whether the GFP is in red/glia cells or not? The separate channels should be shown to see where the glia are. The current picture is neither clear nor convincing

currently.

5. The authors write "As expected, ImpL2 expression levels are reverted to similar levels to the ones observed in controls brains in both cases (fig 1E-F)."

Why is this expected? A priori, ImpL2 may or may not have anything to do with tumor progression. Furthermore, ImpL2 expression could be expected to be cell-autonomously transcriptionally regulated by the activated EGFR or Dp110 in the glia, in which case it would not drop upon knockdown of igloo?

6. Figure 1G: The authors should describe better in the Methods how colocalization is being measured conceptually (besides the software tool that is being used). Is this Mander's colocalization coefficient? Or another one? In particular, is it affected by the absolute signal level of ImpL2? If so, this has more to do with expression levels than colocalization? By eye it does not look like ImpL2 levels upon BSK or Igloo inhibition are staying the same and localizing somewhere else (ie to non-glia)? It appears to me this is mainly quantifying ImpL2 fluorescence. Same with Fig 3G and 4D. In that case, fluorescence levels should be quantified.

7. Likewise, the quantification in Fig 3C needs to be explained better. The legend to Fig 3C simply says "Quantification of pixel intensity", which is obvious by looking at the panel, but it gives no additional information. Is this green/red ratio (which would be most intuitive)? If so, why is the quantification showing a drop when the green mir8 reporter signal is increasing in the glioma sample?

8. The English should be edited by a native speaker or professional service. For instance, in the paragraph starting with "To detect signs of neurodegeneration", the past tense is not written correctly ("we quantify" should be "we quantified", "we knockdown" should be "we knocked down", etc).

9. The authors write "To determine if the increase of ImpL2 in GB depend on miR-8, we analyzed ImpL2 sensor upon miR-8 overexpression in that context. We observed a significant reduction of ImpL2 expression in GB cells in vivo."

Citation of the figure panel is missing.

10. The authors write "Consistently, miR-8 gain-of-function in GB partially rescues the loss of synapses, recapitulating the effect of ImpL2 loss-of function in GB conditions (fig 3D-G)."

The figure citation is wrong. Fig 3D-G does not seem to show or quantify synapses?

11. The panels in Figure 4 are not presented in the figure in the same order as in the text (ie Panel E is referenced before panel A).

August 21, 2020

Dear Andrea Leibfried,

The authors of manuscript #LSA-2020-00693-T have requested an appeal. Their comments are below.

Dear editor,

I am writing this letter to submit our revised version of the manuscript ID#: LSA-2020-00693-T now entitled "Insulin signaling mediates neurodegeneration in glioma" to be considered for publication as a research report in the journal Life Science Alliance.

We have read carefully the comments done by the reviewers and we have followed their suggestions. First we would like to thank the reviewers and editorial board for all the helpful and constructive comments and suggestions on our paper.

We have included further explanations regarding the methods used in the manuscript, with particular attention to the Drosophila specific tools and the analysis of the images.

We have performed all the experiments suggested by the reviewers to reinforce our previous results, we have included extensive discussion about the concerns raised by the reviewers, and we have added new bibliography and new analysis of the results.

As we discussed previously, we have included the analysis of mir-8 binding sites in ImpL2 sequence, a new reporter for Insulin pathway activity in neurons exposed to GB and after ImpL2 RNAi expression in GB cells and quantification of EM images.

Besides, we have tried to generate GB in miR-8 mutant background and combined with miR-8 sponge, but due to the pandemic, flies arrived death due to the delays associated to the COVID19 and we were not able to generate the stocks which required several genetic crosses. Given the lack of strong loss-of-function data, we have softened the affirmations regarding miR-8 as a regulator of ImpL2 expression.

The text has been extensively reviewed and supervised by a professional native speaker

We really hope that this manuscript is now acceptable for publication in LSA

Thank you for your consideration!

Sincerely,

Sergio Casas Tinto, PhD

Researcher, Department of Molecular, Cellular and Developmental Neurobiology

Instituto Cajal, Madrid, Spain

scasas@cajal.csic.es

September 8, 2020

MS: LSA-2020-00693-T

Dr. Sergio Casas-Tintó
Instituto Cajal
Developmental Neurobiology
Avda Doctor Arce, 37
Madrid, Madrid 28002
Spain

Dear Dr. Casas-Tintó,

You appeal for "Insulin signaling mediates neurodegeneration in glioma" [LSA-2020-00693-T] has now been reconsidered, and I am pleased to let you know that we have decided to send your revised manuscript back to the original referees to get their final opinion.

Please use the following link to submit your revised manuscript:

[https://lsa.msubmit.net/cgi-bin/main.plex?
el=A6Na3Sc2A2CMYz3I6B9ftd2PcFKgVsBsh9oOkmOm3AZ](https://lsa.msubmit.net/cgi-bin/main.plex?el=A6Na3Sc2A2CMYz3I6B9ftd2PcFKgVsBsh9oOkmOm3AZ)

Yours sincerely,

Shachi Bhatt, Ph.D.
Executive Editor
Life Science Alliance

Reviewer #1 (Comments to the Authors (Required)):

This is an interesting study of glia-neuron communication signals in the context of glioblastoma. The study reveals the role of *Impl2* in brain tumors and neurodegeneration, which potentially could be further studied to develop anti-tumoral strategies. An issue with the manuscript is that it is unclear whether *miR-8* plays a major role in regulating *Impl2*.

Specific comments:

1. The *miR-8* part of the manuscript needs more supportive data. The authors claim that *miR-8* may regulate *Impl2* indirectly in the discussion. However, to conclude that *miR-8* plays a major role in *Impl2* regulation, the mechanism should be further explored to conclude that *miR-8* regulates *Impl2* expression in GB.

Our results in Figure 3E show that glioma cells upregulate *Impl2* expression, and that *miR-8* overexpression is sufficient to restore *Impl2* expression to control levels (Figure 3G). Therefore, *Impl2* transcription seems to be negatively regulated by *miR-8*. In addition, *Impl2* produced in GB cells mediates the reduction in synapse number in neurons. This synapse reduction is prevented by *miR-8* upregulation in GB cells. All these data prompted us to propose that GB phenotypes mediated by *Impl2* are sensitive to *miR-8* levels.

The simplest explanation would be that *miR-8* bound and regulated directly *Impl2* mRNA levels. We searched for *miR-8* canonical binding sites in *Impl2* mRNA sequence in two widely used databases, STarMir Software for Statistical Folding of Nucleic Acids and Studies of Regulatory RNAs- <http://sfold.wadsworth.org/cgi-bin/index.pl> and TargetSCANFly- <http://www.targetscan.org/>. Unfortunately, we did not find any binding sites indicating a direct *miR-8* regulation of *Impl2*, which suggests an indirect regulation. Finding such mechanism demands a series of experiment that are difficult to address in the current pandemic situation. Besides, despite the interest, we believe that it goes beyond the scope of this manuscript, which is to describe the effect of GB on insulin signaling mediated by *Impl2* and how to prevent it. The molecular mechanism of *miR-8* regulation of *Impl2* expression will be subject of future work from the lab.

We have included further explanation on the functional relation between *miR-8* and *Impl2* in the text to avoid misunderstandings.

2. In addition, there are a number of issues that need to be addressed:
- Figure 3G: I suppose 3G should be quantification of 3DEF and 3L should be quantification of HIJK. In the figure legend, 3G is a quantification of HIJK and 3L is quantification of MNOP which has no active zones.

We have modified the figure legend accordingly.

- The correlation rate between *Impl2*-MIMIC GFP and glial membrane in glioma is less than 2 in figure 1G but is 4 in figure 3G, and controls in these two experiments are same. Why the

difference? If the correlation rate in glioma is 2, the miR-8 expression in glioma (figure 3G) has no significant effect. If the correlation rate in glioma is 4, the sample of figure 1G may have an issue. Or maybe this inconsistency is due to the quantification method?

We thank the reviewer for the observation. Despite we dissected the brains for the different experiments with the same age and at the same moment of the day (12 AM) to diminish variations, the growth of the tumour is highly variable (as it happens in mammals). Due to this variability, we regularly perform specific wt and glioblastoma controls for each experimental condition. In addition, the quantification method requires to set a threshold manually, which could also explain this variability between experiments (for details, see the new, extended part in materials and methods). Each panel (i.e, each set of experiments) has been repeated three times and each one quantified in the same session, in order to avoid as much as possible experimental variability. In addition, all quantifications are done blind.

In the particular case mentioned by the reviewer (correlation rate between Impl2-MIMIC GFP and glial membrane in glioma and control), we always see a statistically significant (although variable) difference between glioma and control. We clarified this in the material and methods session: "Due to the inherent variability of tumor growth and the use of different reporters, we used the appropriate control and glioma genotypes that include them and performed the experiment in parallel for each grouped panel (at least three times): figures 1B-F, 2, 3A-B, 3D-F, 3H-P, 4A-C, 4F-H, 5A-C, 5D-E, 5J-L and 6"

Figures need to be improved:
3. Figure 1C': It is hard to tell where the glial membrane is due to strong GFP. Separate images would be better.

We have modified the figure accordingly.

4. Figure 1G: If some Impl2 staining does not co-localize with glia in gliomas, is Impl2 also expressed in neurons?

Yes, we cannot exclude this possibility. We have used a genomic reporter for Impl2 that monitors the expression of *Impl2* in all cells. It is likely that other cells also express *Impl2* and therefore, they show signal for this reporter.

5. Figure 1: What is the control of E and F? If B serves as control of C-F, the figures should be reorganized. Are these figures (B-F) made from same experiment?

Yes, all these experiments were done in parallel. We have reorganized the figure accordingly.

6. Figure 2F-H: Is H a representative image? It seems that H has dramatically less glial membrane volume compared with control but in quantification (2J) it should be more? In

addition, it is hard to compare the nuclei (green) due to their overlap with membrane (red). Separate images may provide more information, like in figure 3M.

We have modified the figure accordingly, we have separated the green and red channels, and included a more representative image for panel H-H'.

7. All results should be presented in the past tense.

We apologize for this. We have changed the verb tense accordingly.

8. It would be better to use more informative word instead of e.g. "related". Sentences could be more concise and clearer. Here are few examples:

- "In juvenile stages, miR-8 has been related to glial cell growth and positively regulates positively synaptic growth at the neuromuscular junction (26, 27)." -- miR-8 has been found regulates glial cell growth and promotes synaptic growth at the neuromuscular junction...
- "In contrast, *Drosophila* *ImpL2* is related to cachexia, a systemic effect characterized by anorexia and metabolic alterations induced by other malignant tumors (28)". --*Drosophila* *ImpL2* induces cachexia...
- "However, the central function for insulin signaling pathway related to synaptogenesis was described mainly in larval NMJ synapsis (31)." Fix "related to" to "in".
- "Additionally, mitochondrial alterations are related to synapse dysfunction and neurodegeneration (48) (49)" Fix "are related" to "lead".

We have revised the text and made all the suggested changes.

9. Figure 4: Insulin/TOR activity can be detected by an immunostaining of phosphorylated TOR target. This can provide more direct evidence of TOR activity and is a good addition to the transcriptional changes.

We have tried phosphoTOR antibody with very little success. However, we took an alternative way and used a transgene (PH-GFP) that, using a pleckstrin homology domain –green fluorescent protein fusion, reports the activity of PI3K and thus, is used as a monitor of Insulin pathway (Figure 4). (Britton et al, Dev Cell 2002; doi [10.1016/S1534-5807\(02\)00117-X](https://doi.org/10.1016/S1534-5807(02)00117-X)).

The results show that PH reporter activity in neurons is reduced upon GB induction, in addition, *ImpL2* knockdown in GB prevents this phenotype, and the PH-GFP reporter expression is restored to normal levels. These results indicate that *ImpL2* expressed in GB causes an attenuation of Insulin signalling pathway in neurons.

10. Figure 4E: Rp49 is not a proper housekeeping gene as activated insulin signaling/

overexpression of Rheb can lead upregulation of ribosomal proteins. Figure 4E may not reflect to the actual levels of Rheb. Same issue in Figure 1A - although *Impl2* is likely upregulated.

All the reviewers have raised concerns about this particular experiment (qPCR to measure *Rheb* mRNA). There is a lack of information in the literature to support that *Rheb* transcription is specifically regulated by Insulin pathway activity in wt conditions. Our results regarding *Rheb* transcription, by themselves, are not conclusive and in view of the new results included with the PH-GFP reporter, do not contribute to the main message of the manuscript. Therefore, we have removed *Rheb* qPCR results in the resubmitted version except to show that the LexAOp-dRheb tool works.

Regarding the use of *rp49*, we validated *rp49* expression levels and compared control, GB and GB + Rheb samples per triplicate. In our hands, *rp49* expression levels are similar in all samples and this was the best available housekeeping gene as compared with RNAPolIII or Actin. Besides, *Impl2* upregulation results (qPCR) are validated with the increase of signal of *Impl2* reporter data (Figure 1) in GB conditions. Therefore, the main message from both experiments suggests an upregulation of *Impl2* in GB.

Finally, regarding Insulin pathway activity in neurons, we have included new data in Figure 4 that now reports Insulin pathway attenuation in neurons using PH-GFP reporter, and Thor-MIMIC reporter. As well, the neuronal phenotypes (synapse number) caused by the GB, or *Impl2* expression in glial cells, is reproduced by the genetic inhibition or the insulin receptor (Figure 4H-J) in neurons.

11. Figure 5 J-L: Conclusion of these figures can only be made with a quantification of the phenotypes.

We have quantified the results from EM experiments and included these quantifications in Figure 5N.

12. Figure 5: Insulin signaling has broad effects. Overexpression of Rheb completely reverted effects of glioma (5GHI), whereas the mitochondrial alterations seem to be only partially rescued. I wonder if some other functions of insulin signaling is critical for the phenotype of glioma but not the mitochondrial physiology.

This is a very interesting point for discussion and we agree with the reviewer. It is difficult to separate the phenotype of glioma from the mitochondrial physiology, as this is directly related to synapse number which is also reduced as a consequence of GB expansion. We described recently the competition between GB and neurons for Wingless (WNT), and the consequences in neurons of this Wg depletion (Portela et al, PLOS Biology; doi: [10.1371/journal.pbio.3000545](https://doi.org/10.1371/journal.pbio.3000545)).

We propose here that GB cells impact on healthy neuron physiology through *Impl2* and Insulin pathway. Our results show that upregulation of *Rheb* (Insulin pathway) in neurons fully rescues

neuronal associated phenotypes, including the accumulation of mitochondria in projections and NMJ (5H and 5I). However, *Rheb* upregulation in neurons exposed to GB not only rescues the number of active zones but also causes a significant increase compared to control samples. We recently described molecular mechanisms that change synapse number and PI3K pathway plays a central role in synapse number regulation, so it is reasonable that *Rheb* causes this phenotype (Portela et al, PLOS Biology; doi: [10.1371/journal.pbio.3000545](https://doi.org/10.1371/journal.pbio.3000545)).

It is true that *Rheb* overexpression only partially rescues mitochondrial morphology but on the other side it rescues life span experiments. Mitochondrial biology (morphology and number of mitochondria, fusion/fission events) is very sensitive to different inputs, including Insulin pathway as it is a potent driver of cellular growth and metabolism. Reduced insulin signalling leads to lower expression of genes encoding mitochondrial genes (Gershman et al, Physiol Genomics 2007 doi: [10.1152/physiolgenomics.00061.2006](https://doi.org/10.1152/physiolgenomics.00061.2006) and Teleman et al, Cell Metab 2008; doi: [10.1016/j.cmet.2007.11.010](https://doi.org/10.1016/j.cmet.2007.11.010)). In addition, Insulin stimulates mitochondrial fusion and fission in cardiomyocytes (Parra et al, Diabetes 2014; doi: [10.2337/db13-0340](https://doi.org/10.2337/db13-0340)).

So, the morphology of the mitochondria rescued by *Rheb* expression could be a consequence of overstimulation of the Insulin pathway, or, on the other hand, it might be the minimal mitochondrial "normal" morphology required for functional rescue. Clearly, further studies on mitochondrial activity are required to better understand the impact of GB in neurons.

Minor Issues (in order of appearance in the manuscript):

1. Line and page numbers would be helpful for the review process.

We have included line and page numeration.

2. Abstract: "Therefore, signals from GB to neuron emerge..." should be fixed to "Therefore, signals from glioblastoma to neuron emerge..." or start using GB by the first "glioblastoma" word.

We have changed this in the abstract.

3. Introduction: "This model is based on two of the most frequent mutations in patients, a constitutively active form of the epidermal growth factor receptor (dEGFR λ) and the phosphatidylinositol-3 kinase (PI3K) catalytic subunit p110 α (PI3K92E) driven by the glial specific repo-Gal4 (16)", dEGFR λ should be EGFR.

We have corrected this accordingly.

4. "in GB development, metastasis, therapeutic response, and prognosis (reviewed by (21))." - "and prognosis (reviewed in 21)"

We have changed this.

5. "We have recently re-evaluated GB as a neurodegenerative disease, showing that GB reduces the number of synapses through wingless/frizzled 1 (*wg/fz1*) signaling (Portela et al, PLOS Biol 2019), equivalent to mammalian WNT pathway (36)". The reference should be fixed. Same issue in the first paragraph of results section.

We have corrected the reference.

6. "However, whether tumoral glial cells are able to modify insulin signaling directly in neurons, and consequently alter the number of synapses, is yet unknown." I suppose the "directly" in this sentence should be "remotely"?

Thank you very much for this suggestion, we have corrected the sentence.

7. In figure legends: "***p-value>0,005, ***p-value>0,0001" should be p-value<0,005 and p-value<0,0001. Same mistakes can be found in other figures.

We have changed the symbol in all figure legends.

8. "Consistently, GB cells show higher GFP levels than control glial cells. Likewise, upon *ImpL2* RNAi expression we detect a decrease in GFP levels, similar to the ones observed in control brains (fig 1B-D)." This sentence needs rephrase.

We have rephrased the sentence as follows:

To discriminate *ImpL2* expression in neuronal or glial (GB) cells, we used a MIMIC GFP reporter that reproduced faithfully *ImpL2* expression. Consistently, GB cells showed higher reporter GFP levels than control glial cells, which are restored to control levels upon *ImpL2* knockdown (fig 1B-D).

9. Figure 1 G: It seems that the N is more than 10 in each genotype as shown by the number of dots in the figure. Should be fixed to the correct number. Same issue in Figure 2E, 3G, 3L, etc.

We have corrected this in the manuscript. The dots correspond to the number of measurements meanwhile the N is the biological number of samples used in the experiments. Except in the case of quantification of total number of glial nuclei, it is possible to obtain more than one data from the same sample.

10. Figure 2: In figure title, "*ImpL2* downregulation in glioma cells causes neurodegeneration

and reduces tumor progression". Isn't that Impl2 knockdown counteracted neurodegeneration?

We have changed this in the figure legend.

11. "The GB itself is induced by overexpressing a constitutively active form of PI3K, thus the insulin pathway is activated in all glial cells. However, mRNA levels of dRheb are reduced in GB brains when compared with control brains, suggesting that this increase reflects mostly neuronal expression (fig 4E)." This conclusion is not clear to me, what is "this increase" in neuron?

As explained above, we have removed *Rheb* qPCR data from the manuscript and included further evidences of Insulin pathway activation in neurons (Figure 4).

12. Discussion: "GB is the most aggressive type of brain tumor." The word "most" is too strong, I suggest rewording to "one of the most".

We have modified this sentence accordingly.

We thank the reviewer for his/her comments on minor issues. We have gone through the text and changed them as suggested.

Reviewer #2 (Comments to the Authors (Required)):

Insulin signaling mediates neurodegeneration in glioma
Patricia Jarabo, Carmen de Pablo, Héctor Herranz, Francisco Antonio Martín and Sergio Casas-Tintó¹

In this interesting, and well-written, study, Jarabo and colleagues authors show that the secrete Impl2 signal from glioblastoma-like cells dampens insulin signalling leading to neurodegeneration accompanied or caused by mitochondria alterations. The authors also show that overexpression of dRheb, a gene that was significantly downregulated in GB brain, specifically in neurons could rescue neuronal degeneration caused by glioblastoma cells and, more strikingly, suppressed tumorigenesis and rescued GB-mediated lethality.

Most of the conclusions are well supported and the figures are of quality. Moreover, if the mechanism is shown to be conserved in other animal species such as mice, the findings may open a new direction to study the glioblastoma-microenvironment interactions and the fly GB model could be of utility to future studies to integrate the numerous pathways affecting the in vivo invasion of GB cells.

I have some questions/comments or concerns to the authors:

(1) Page 6. ImpL2 mediates GB progression and neurodegeneration Figure 2 A-C, E. Here the authors cannot distinguish between the effects of ImpL2 on tumorigenesis or those non-autonomous on neurodegeneration.? ImpL2 downregulation reduced glioma cell membrane expansion, a feature of GB. However, later, they show that in wt glial cells overexpression of ImpL2 alters the number of synapses. While the data are a correlation, the authors could present the data indicating that together the most parsimonious explanation is that ImpL2 impacts both tumorigenesis and neurodegeneration.

Note that this is different to the situation in the cancer-cachexia models

This is a very interesting observation and a matter for further discussion. It is intriguing if ImpL2 impacts independently both tumor expansion (GB membrane) and neurodegeneration or if these two events are dependent. We have recently described that specific properties of neurons (like quantity of Frizzled receptor), can modulate GB expansion (Portela et al, PLOS Biology; doi: [10.1371/journal.pbio.3000545](https://doi.org/10.1371/journal.pbio.3000545)). Moreover, in this manuscript we show that *Rheb* upregulation in neurons not only rescues cellular and functional features altered by GB but also hampers GB progression (Figure 6). We have now included this discussion in the manuscript to propose both possibilities, the double role for ImpL2 on the bidirectional communication between GB cells and healthy surrounding neurons.

(2) Figure 2. "Downregulation of ImpL2 causes neurodegeneration..." Shouldn't it state that the opposite? ImpL2-RNAi rescued the number of active zones and thus rescues neurodegeneration? Indeed, in results the authors state " The results show that ImpL2 reduction in GB cells counteracted the reduction in the number of synapses-synapses of GB brains"

We have corrected this error in the Figure legend.

(3) MicroRNAs regulates... It should say miRNA regulated

We changed microRNA to miRNA throughout the text.

(4) miR-8-ImpL2. This needs further validation. Authors should more directly measure the levels of mir-8. While the ability of overexpression of mir-8, a known regulator of ImpL2, has an impact, this alone is not sufficient evidence of the contribution of mir-8 in GB. a. Analysis using either SP-mir-8, or epistasis using mir-8 mutations should be added to corroborate this conclusion. It would be expected that depletion of mir-8 would further increase ImpL2 levels, connecting ImpL2 to endogenous mir-8.

We agree with the reviewer. In fact, we have tried to generate GB in *miR-8* mutant background and combined with *miR-8* sponge, but due to the pandemic, flies arrived death and it has been impossible to build on the stock in time. In consequence, and given the lack of strong loss-of-function data, we have softened the affirmations regarding *miR-8* as a regulator of ImpL2. We deeply apologize with the reviewer.

(5) I have a question about the quantifications of miR-8 experiments. When comparing all graphs of Active zones, I noted that control in the miR-8 experiment show ~1500 active zones, whereas in the other graph is ~1000 active zone. This is a significant difference which questions whether the validity of the conclusions based on this graph? Could the authors elaborate on this discrepancy in the numbers in the various controls.

We are aware of these variances in the number of synapses. Indeed, slight changes in temperature, light or the moment of the dissection can alter synapse number. In consequence, every experiment is accompanied by its own control performed in parallel at the same moment of the day (12 AM), and all the experiments were done three times blind. To compare results among different experiments, it is widely extended to normalize with control samples, but we decided to present the data in this format because we believe it is more informative regarding the intrinsic variability of the experimental procedure. However, if required we could change all the graphs and normalize the data to control.

(6) Is the data representing the same genotypes in the different figures? If so, this should be explained

We thank the reviewer for giving us the opportunity to clarify this issue. Despite we dissected the brains with the same age and at the same moment of the day (12 AM) to diminish variations, the growth of the tumour is highly variable (as it happens in mammals). Due to this variability, we regularly perform specific wt and glioblastoma controls for each experimental condition. In all the figures the genotype of control and glioblastoma animals is the same but coming from experiments performed in slightly different conditions. In addition to this, in the experiments with reporters (like figures 1A-F, 3A-B, 3D-F, 4A-C and 5D), the genotype of control and glioblastoma animals also included the reporter, to diminish experimental differences. This is annotated in the figure legends. We clarified this in the material and methods session: "Due to the inherent variability of tumor growth and the use of different reporters, we used the appropriate control and glioma genotypes that include them and performed the experiment in parallel for each grouped panel (at least three times): figures 1B-F, 2, 3A-B, 3D-F, 3H-P, 4A-C, 4F-H, 5A-C, 5D-E, 5J-L and 6".

(7) GB secreted ImpL2 reduces neuronal Insulin signaling
The authors state " To evaluate the impact of insulin signalling reduction in neurons, we measure dRheb mRNA by qPCR. dRheb is the molecular link between insulin signalling and TOR kinase, and it reflects the insulin pathway activity (reviewed in 42)"

Nothing in this review supports this statement. Insulin and nutrient regulation of Rheb/mTORC1 signaling relies on activation of Rheb via sub cellular location not transcription. What is the evidence that mRNA of Rheb reflects IIS activity? Saucedo (2003) has shown that mRNA Rheb is elevated in protein starved animals, but not in fed animals. This result does not support, by itself, that levels of mRNA Rheb is a proxy of IIS activity

This issue has been raised by the three reviewers. Their comments about measuring *dRheb* mRNA by qPCR are totally fair and we understand their concerns. First of all, we apologize for the mistake in the quoted paper. As the reviewer highlights, we could not find any reference showing that *Rheb* transcription is specifically regulated by Insulin pathway activity in wt conditions.

Our results regarding *Rheb* transcription, by themselves, are not conclusive and in view of the new results included with the PH-GFP reporter (see below), do not contribute to the main message of the manuscript. Therefore, we have removed *Rheb* qPCR results in the resubmitted version. We included qPCR to show that the LexAOp-*Rheb* transgene is functional and produces an increase of *Rheb* mRNA (Fig 5A).

Finally, regarding Insulin pathway activity in neurons, we have included new data in Figure 4 that now reports Insulin pathway attenuation in neurons using PH-GFP reporter, and Thor-MIMIC reporter. As well, the neuronal phenotypes (synapse number) caused by the GB, or ImpL2 expression in glial cells, is reproduced by the genetic inhibition or the insulin receptor (Figure 4H-J) in neurons.

(8) Page 9, the authors state " However, mRNA levels of dRheb are reduced in GB brains when compared with control brains, suggesting that this increase (?) reflects mostly neuronal expression (fig. 4E). What increase? If levels are decreased. What is the evidence that the change reflects mostly the neuronal expression?

We have included new data (PH-GFP) to specifically monitor Insulin pathway activity, and we have corrected the text accordingly.

If Rheb mRNA levels are inversely correlated with amino acid levels (Saucedo, 2003), it would be expected to be also inversely correlated to IIS activity. The observation that levels are reduced does not support the authors' claim of IIS reduction in GB brain neurons. More, the authors must explain what evidence supports that the observed reduction of mRNA of dRheb in GB brains is brought about by ImpL2?
(9) Transcriptional regulation of THOR reflects dFOXO activity, not dTOR control because dTOR regulates THOR protein by phosphorylation and this is not examined here.

We are sorry for this conceptual error. We changed it in the manuscript and included FOXO as the transcriptional regulator of *Thor*.

(10) The authors state: "neurons confronted with GB cells have reduced insulin signaling". Since the images in Figure 4 only shows the positive dots of THOR-MiMIC with respect to Elav, it seems appropriate to eliminate "confronted" of the text.

We have corrected this in the text as suggested: "Neurons exposed to GB...".

(11) 'All these results together suggest that *ImpL2* up-regulation in GB cells mediates the decreased insulin pathway activity detected in neurons" This is an unnecessary overstatement. The data are suggestive of a likely paper of IIS in neurons, and forcing conclusions by over-interpreting does not help. Moreover, the status of IIS in neurons need to be confirmed more convincingly.

We agree with the reviewer; the sentence is too strong given our current data. We have modified the text as follows: "All these results together suggested that *ImpL2* up-regulation in GB cells might decrease the activity of insulin pathway in neurons, which might cause neurodegeneration".

(12) Images in Figure 5D-F and D'-F' have no resolution to see single mitochondria and to make any conclusion. Can the authors explain in what sense one would expect that the increased fluorescence intensity in the Cherry-mito GB brain to reflects neurodegeneration?

Mitochondria integrity alterations, changes in morphology, aberrant distribution and accumulation are features of neurodegeneration (for instance, see Debattisti and Scorrano, *Molecular and Cellular Neuroscience* 2013; doi: [10.1016/j.mcn.2012.08.007](https://doi.org/10.1016/j.mcn.2012.08.007) and Deal and Yamamoto, *Front Genet* 2018; doi: [10.3389/fgene.2018.00700](https://doi.org/10.3389/fgene.2018.00700)). Also, cherry-mito constructs have been previously used to visualize mitochondria dynamics in neurodegeneration (Vagnoni and Bullock, *Nat Protocols* 2018; doi: [10.1038/nprot.2016.112](https://doi.org/10.1038/nprot.2016.112)) and accumulation of mitochondria is characteristic of neurodegeneration processes (Sterky et al 2011, *PNAS* [10.1073/pnas.1103295108](https://doi.org/10.1073/pnas.1103295108)).

(13) Figure 5J-L We need here quantification, given that the analysis with fluorescence Cherry-mito did not yield sufficient resolution. The image of dRheb brain is unconvincing. The number of brains / cases of defective mitochondria and 'rescue' has not been included.

We have quantified the EM images and included the number of brains. The average size of GB mitochondria is significantly smaller compared to control. Overexpressing *Rheb* in the neuronal population of flies with GB rescues this phenotype, restoring the mitochondrial size to control levels. However, we agree that the morphology of mitochondria is partially rescued and the consequences of these phenotypes, should be studied in depth, but this is out of the scope of this manuscript and of our expertise.

(14) Figure 6. Glioma elav>Rheb the size of this brain is almost as control. This is a rather intriguing observation which should be further supported.

We agree with the reviewer. As indicated in panels D and E, *Rheb* overexpression in neurons also affects the increase in GB cell number, and the total volume of the GB. This is now included in the discussion, i.e. how the interaction between GB cells and healthy surrounding cells is relevant for the expansion of the tumor. In line with these results and previous publications from the lab, we are currently studying the impact of “neuronal health” in the progression of GB and the idea of “strong” brains that resist better the presence of a GB. Indeed, we showed that other modification in the neurons (Frizzled 1) not only protects neurons from the GB, but also affect the propagation of the tumor experiments (Portela et al, PLOS Biology; doi: [10.1371/journal.pbio.3000545](https://doi.org/10.1371/journal.pbio.3000545)). Now here, the experiment with *Rheb* supports this idea.

Ideally, IIS should be manipulated more directly via Pi3k/Akt/Pten in neurons and the status of pAkt and not mRNA of Rheb be assessed.

The reviewer is right. Unfortunately, there are no tools for LexA modulating PI3K, PTEN or Akt that we are aware of. We generated LexAop-Rheb for this manuscript and validated the tool properly (Fig 5A). Regarding the Rheb mRNA, the three reviewers raised concerns about the experiment. Actually, we could not find any reference showing that *Rheb* transcription is specifically regulated by Insulin pathway activity in wt conditions. We agree with the reviewer that this experiment is controversial so we removed it in the resubmitted version regarding insulin pathway activity. It is included to prove that LexAOp-*dRheb* tool is functional.

However, we have included further experimental evidences to assess the status of AKT. We used a transgene (PH-GFP composed by the fusion of a pleckstrin homology domain plus Green Fluorescent Protein) that reports the activity of PI3K and thus, is widely used as a monitor of Insulin pathway (Britton et al, Dev Cell 2002; doi [10.1016/S1534-5807\(02\)00117-X](https://doi.org/10.1016/S1534-5807(02)00117-X)). These new results show that PH reporter activity in neurons and glia is strongly reduced upon GB induction. In addition, *ImpL2* knockdown specifically in GB prevents this reduction (compare fig 4A with 4C) both in neurons and glia, indicating activity in neurons (not repo) of the insulin signalling pathway.

Discussion:

(15) Finally, here we also describe a one-way communication system from GB cells towards neurons.

This statement is probably incorrect because the manipulation of IIS via *dRheb* in neurons suppressed GB progression suggesting that communication is bi-directional as seen previously by others.

Thank you very much for this comment, we agree with the reviewer and we have corrected the text.

(16) *ImpL2* binds DILPs and this may reduce insulin signaling in neurons. The rescue of 'mitochondrial aberration' is not convincing.

We have included further evidences on the activity of the insulin pathway in neurons and we have quantified the rescue of mitochondrial defects.

(17) We have described the presence and relevance of miR-8 in GB progression as a regulator of ImpL2 expression. This conclusion is based on correlative data and as such should be described in that way. Epistatic analysis could support and verify this idea.

We agree with the reviewer. In fact, we have tried to generate GB in *miR-8* mutant background and combined with miR-8 sponge but due to the pandemic, flies arrived death and it has been impossible to build on the stock in time. In consequence, and given the lack of strong loss-of-function data, we have softened our affirmations regarding *miR-8* as a regulator of *ImpL2*. We apologize with the reviewer.

(18) Authors should discuss in an inclusive way the potential relationship between neurodegeneration by WG/WNT and IIS

This is a very interesting observation and we have included it in the discussion.

Minor:

- o 'the aim is not to heal a sick insect' I feel that this sentence is unnecessary. Those who might think this way are unlikely to be readers of this study
- o In Page 5. Portela et al 2019, eliminate PLOS Biol
- o Methods eliminate UAS- in ImpL2-MI14001
- o As far as I know all available mir-8-sensors are driven by the tubulin promoter not the UAS. And this include the sensor in reference 41. I could be wrong, but I suggest the authors to check this, too.
- o Fig. 5G. The order of this figure should be rearranged panel G is discussed before D
- o It should be corrected as Cherry-mito because in this construct is the N-terminus that is tagged with mCherry.
- o Please, add the citations to the statements on human GB in the first sentences of the Discussion
- o The titles shouldn't say something like: Reduced Insulin Signaling Mediates ...

We thank the reviewer and changed the text following his/her suggestions.

Reviewer #3 (Comments to the Authors (Required)):

The authors propose here that glia overexpressing activated EGFR and PI3K secrete the insulin antagonist ImpL2, leading to reduced insulin signaling in neurons, and thereby phenotypes such as tumor progression and reduced neuronal synapses.

Overall, this is an interesting story. However, as detailed in 'major issues' below, several key findings are not solid, and some results don't seem to fit together. Furthermore, the manuscript is put together in a very sloppy way, with incorrect citations, lacking figure citations, and a lack of information on how assays were done in the figure legends and M&M, which makes it difficult to follow and to assess. These major issues should be addressed prior to publication.

Major

Issues:

1. If activation of EGFR and PI3K in glia leads to cell-autonomous inhibition of mir8 expression and thereby induced *ImpL2* levels, why do knockdown of basket or igloo in glia block this effect? Just because these two genes are required for formation of tumor microtubules, and thereby tumor progression, doesn't mean they should block the cell autonomous signaling pathway from EGFR+Dp110CAAX to mir8 to *ImpL2* ? Is mir8 regulated cell autonomously by JNK signaling? If so, why? Do tumor microtubules regulate mir8 expression somehow? How?

The potential relationship between WG/WNT and Insulin pathway has been proposed under physiological or tumoral conditions (Yi et al, *Endocrinology* 2008; doi:[10.1210/en.2007-1142](https://doi.org/10.1210/en.2007-1142), and Desbois-Mouthon et al, *Oncogene* 2001; doi: [10.1038/sj.onc.1204064](https://doi.org/10.1038/sj.onc.1204064)) and represent a potential issue of interest to study in GB-host biology. We described recently (Portela et al, *PLOS Biology*; doi: [10.1371/journal.pbio.3000545](https://doi.org/10.1371/journal.pbio.3000545)) the positive feedback loop established with wingless/JNK/MMPs and tumour microtubules that promote GB progression. We do not have evidences that *miR-8/ImpL2* regulation is directly controlled by EGFR and/or PI3K signalling pathways, however, the results included in Figure 1 suggest that a reduction of JNK pathway (*BSK^{DN}*) or the knockdown of *igloo* (prevention of TMs formation) reduces *ImpL2* expression. Our data suggest that *ImpL2* upregulation is sensitive to TMs formation and JNK, and one could speculate that it might be also dependent on Wg/WNT signalling pathway. However, our observations in *Drosophila* suggest that both pathways (Insulin and Wg) participate in the equilibrium between GB cells and neurons.

About a possible link between JNK and *miR-8*, it has been described that *miR-8* mutant animals activate JNK signaling, but there is no evidence that JNK can regulate *miR-8*. However, miRNA regulation has the tendency to establish reciprocal feedback loops and networks (Herranz and Cohen, *Genes Dev* 2010; doi: [10.1101/gad.1937010](https://doi.org/10.1101/gad.1937010)), so it might be plausible that JNK signaling and *miR-8* would have such a reciprocal regulation. The relations among all different pathways and the mutual regulation should be matter of study of future projects. We have included this discussion in the manuscript.

2. According to the model, inhibition of mir8 by EGFR+Dp110CAAX leads to increased Impl2 levels and thereby 3 phenotypes: reduced synapses, increased glial cell number, and increased glial cell membrane volume. Indeed, knockdown of Impl2 rescues these three phenotypes (Fig 2). Mir8 overexpression seems to efficiently block the increase in Impl2 levels in the tumors (Fig 3G). Then why doesn't it revert all the phenotypes - ie not glial cell number? The authors propose that mir8 overexpression also regulates some other gene that promotes glial proliferation. That's possible, but surprising. Unfortunately, from the materials & methods it is not clear how exactly glial cell number was measured/quantified. Were the tissues stained with a nuclear marker, which was then segmented and quantified, or was the glial membrane marker used to somehow count cell number? (The latter seems to be the case from the description). If so, how can changes in cell number be distinguished from glia that just have more or less membrane? In sum, it is not clear whether glial cell number is quantified correctly, and hence whether these conclusions are solid.

Glial network was marked by a UAS-myristoylated-RFP reporter specifically expressed under the control of *repo*-Gal4. The total volume was quantified using Imaris surface tool (Imaris 6.3.1 software). Glial nuclei were marked by staining with the anti-Repo (DSHB-8D12) recognizing specifically glial nuclei. The number of Repo+ cells was quantified by using the spots tool of Imaris 6.3.1 software. We selected a minimum size and threshold for the spot in control samples for each experiment. Then we applied these conditions to the analysis of each corresponding experimental sample. We have explained this in the Materials and Methods section.

The specific targets of *miR-8* are not known, but the results included in figure 3 show that the *miR-8* upregulation in glial cells is sufficient to cause a significant increase of glial cell number, but not of the volume of these cells, as it does in a GB condition. Thus, we propose that *miR-8* plays a role in the establishment of glial cell number.

3. It is not clear whether the mir8 overexpression experiments (Fig 3) are overexpressing mir8 to physiological levels (ie restoring it to wildtype glia levels) or to very high supraphysiological levels. This could be quantified, to show that the results are physiologically relevant. Alternatively, since mir8 null mutants are viable, the contribution of mir8 can be rigorously tested by measuring the phenotypic readouts (eg active zones) in a mir8 knockout background (ie compare mir8KO versus mir8, *repo*>EGFR+dp110CAAX).

This is an interesting issue. Actually, the overexpression of *UAS-miR-8* in the larval fat body rescued the body weight and size to near wt levels (Hyon et al, Cell 2009: doi: [10.1016/j.cell.2009.11.020](https://doi.org/10.1016/j.cell.2009.11.020)), suggesting that the same *UAS-miR-8* tool that we have used does not produce very high levels of *miR-8* but instead near to physiological levels.

4. The drop in insulin signaling levels in neurons is not solid: -The authors write "dRheb mRNA levels drop down when Insulin signaling is low (43)." but I did not find this in the cited reference.

-The Rheb Q-RT-PCR result (Fig 4E) is strange. Presumably, according to the authors' interpretation, it should be increasing in the glia where Dp110 is activated, and decreasing in the neurons as a result of the secreted ImpL2 which reduces insulin signaling. However, overall, Rheb mRNA levels seem to be dropping to less than 5%. How can that be? Here is one example (besides the ones listed below) of a poorly described experiment. In the figure legend, it says "RT-qPCR of Rheb expression is downregulated in gliomas". Is only the glioma being dissected out for the assay? I assume not (it would probably require laser capture), and that RNA is being extracted from the entire brain? In which case these are Rheb levels in the brain, not in the glioma. In that case, is only 5% of the brain composed of glia? This seems unlikely. Even in that case, it would require Rheb mRNA levels to drop essentially to zero in the neurons. How do the authors know that the Rheb they are measuring is neuronal Rheb and not glial Rheb?

This issue has been raised by the three reviewers. Their comments about measuring *dRheb* mRNA by qPCR are totally fair and we understand their concerns. First of all, we apologize for the mistake in the quoted paper. As the reviewer highlights, there is a lack of literature supporting that *Rheb* transcription is specifically regulated by Insulin pathway activity in wt conditions. The RNA was extracted from the entire brain, so we cannot rule out the glial contribution, despite that in wt brains (but not brains with GB) glia represents up to 10% of total cell population in the *Drosophila* adult brain (Freeman, CSH Perspective 2015; doi: 10.1101/cshperspect.a020552). Actually, our results regarding *Rheb* transcription, by themselves, do not prove that Insulin signalling is altered in GB conditions or is modified by ImpL2. In summary, we agree with the reviewer that this experiment is controversial so we removed it in the resubmitted version, except to show that LexAOp-*Rheb* transgene expresses *Rheb* effectively.

In addition, we have included new further data on the activity of the Insulin pathway in neurons using an additional reporter (PH-GFP) (Figure 4).

-Fig 4A-C: it is not clear what is being imaged/analyzed here. Are these neurons or glia? According to the author's interpretation, insulin signaling should be high in glia (due to *repo>Dp110CAAX*) and low in neurons. Hence it makes a difference if Thor levels are being observed in neurons or glia. However there are no markers for neurons or glia in the image, and it is not even clear which region of the brain is being analyzed.

We have included further details of the quantification protocol in Materials and Methods. In this particular image, we have used an anti-elav antibody (blue) that is specific for neurons (DSHB). In addition, the new result of PH-GFH support the result obtained for Thor.

-One way to solidify this would be to look at a GFP-based PtdIns(3,4,5)P3 sensor, which has cellular resolution, and hence can be combined with glial/neuronal markers.

We agree with the reviewer and we performed the experiment as he/she suggested. These

new results show that PH reporter activity in neurons is strongly reduced upon GB induction (fig 4 A-C). In addition, *Impl2* knockdown specifically in GB prevents Insulin pathway signalling reduction in neurons.

Given that *Impl2* was originally described a secreted antagonist of the insulin pathway (Honegger et al, J Biol 2008; doi: [10.1186/jbiol72](https://doi.org/10.1186/jbiol72)), high levels of *Impl2* should decrease the synapse number in neurons, as the neuronal down-regulation of Insulin signalling does (*InR^{DN}*) (fig 4F-I). All these data support the hypothesis that *Impl2* expressed in GB causes the attenuation of Insulin pathway in neurons.

Minor

Issues:

1. The abstract is misleading as currently written. The authors write "Here we describe how the glioblastoma produce *Impl2*, an antagonist of the insulin pathway, which is regulated by the microRNA miR-8."

It should be made clear in the abstract that what is being studied is a *Drosophila* model for glioblastoma, and not actual glioblastoma.

We apologize for not mentioning *Drosophila* in the abstract, our mistake. We made clear that the work is done in a *Drosophila* glioblastoma model that reproduces faithfully features of human glioblastoma.

2. The first paragraph of the Results section is actually still introduction.

We moved this to the last part of the introduction.

3. Legend to Figure 1: -Legend to 1A should indicate whether the glioma is being dissected out (as suggested by the legend "upregulation of *Impl2* in gliomas") or whether this Q-RT-PCR is being done on whole brains or whole heads? -"B)" is missing at the beginning of the description of panel B.

We have included the required information in the figure legend.

4. In Figure 1C-C' I cannot see whether the GFP is in red/glia cells or not? The separate channels should be shown to see where the glia are. The current picture is neither clear nor convincing currently.

We have split the channels as suggested.

5. The authors write "As expected, *Impl2* expression levels are reverted to similar levels to the ones observed in controls brains in both cases (fig 1E-F)." Why is this expected? A priori, *Impl2* may or may not have anything to do with tumor progression. Furthermore, *Impl2* expression could be expected to be cell-autonomously

transcriptionally regulated by the activated EGFR or Dp110 in the glia, in which case it would not drop upon knockdown of igloo?

The reviewer is right. This was not an expected result and the alternative possibility that he/she suggested might have been correct. We did not anticipate his/her conclusion, so if the reviewer does not mind we would like to add it. So we have removed "as expected" and adding indicating that glial *ImpL2* expression was not transcriptionally regulated by activated EGFR or Dp110".

6. Figure 1G: The authors should describe better in the Methods how colocalization is being measured conceptually (besides the software tool that is being used). Is this Mander's colocalization coefficient? Or another one? In particular, is it affected by the absolute signal level of *ImpL2*? If so, this has more to do with expression levels than colocalization? By eye it does not look like *ImpL2* levels upon BSK or Igloo inhibition are staying the same and localizing somewhere else (ie to non-glia)? It appears to me this is mainly quantifying *ImpL2* fluorescence. Same with Fig 3G and 4D. In that case, fluorescence levels should be quantified.

We have included further description in the materials and methods section.

7. Likewise, the quantification in Fig 3C needs to be explained better. The legend to Fig 3C simply says "Quantification of pixel intensity", which is obvious by looking at the panel, but it gives no additional information. Is this green/red ratio (which would be most intuitive)? If so, why is the quantification showing a drop when the green mir8 reporter signal is increasing in the glioma sample?

We have included further description in the materials and methods section.

8. The English should be edited by a native speaker or professional service. For instance, in the paragraph starting with "To detect signs of neurodegeneration", the past tense is not written correctly ("we quantify" should be "we quantified", "we knockdown" should be "we knocked down", etc).

The English has been reviewed by an expert.

9. The authors write "To determine if the increase of *ImpL2* in GB depend on miR-8, we analyzed *ImpL2* sensor upon miR-8 overexpression in that context. We observed a significant reduction of *ImpL2* expression in GB cells in vivo." Citation of the figure panel is missing.

We have corrected this in the text.

10. The authors write "Consistently, miR-8 gain-of-function in GB partially rescues the loss of synapses, recapitulating the effect of Impl2 loss-of function in GB conditions (fig 3D-G)." The figure citation is wrong. Fig 3D-G does not seem to show or quantify synapses?

We have corrected this mistake. Now the figure citation is 3H-J, L.

11. The panels in Figure 4 are not presented in the figure in the same order as in the text (ie Panel E is referenced before panel A).

We have modified this in the figure and text.

.....

January 4, 2021

RE: Life Science Alliance Manuscript #LSA-2020-00693-TR-A

Dr. Sergio Casas-Tintó
Instituto Cajal
Developmental Neurobiology
Avda Doctor Arce, 37
Madrid, Madrid 28002
Spain

Dear Dr. Casas-Tintó,

Thank you for submitting your revised manuscript entitled "Insulin signaling mediates neurodegeneration in glioma". We would be happy to publish your paper in Life Science Alliance pending final revisions to address the minor concerns raised by Reviewer 2, including the points about Figure 4 and 5, and additional revisions necessary to meet our formatting guidelines.

Along with the points listed below, please also attend to the following,

- please add your figure legends to the main manuscript text
- Figure 4 legend is missing panel J,K; is Figure 3 legend missing Panel I - please add
- please either as dotted insets to show the area being zoomed in Figure 4D',E',F' OR clarify in the legend that the insets shown in Figure 4D,E,F show the same area that is also shown zoomed in in Figure 4D',E',F'
- please try to make the scale bars more visible in Fig. 5 K,L,M; Fig. 3A,B

A. FINAL FILES:

B. MANUSCRIPT ORGANIZATION AND FORMATTING:

Sincerely,

Shachi Bhatt, Ph.D.
Executive Editor
Life Science Alliance
<https://www.lsjournal.org/>
Tweet @SciBhatt @LSAJournal

Reviewer #1 (Comments to the Authors (Required)):

The issues raised in my review were addressed by the authors. The additional experiments performed and the re-writing of some sections better represent the novelty and importance of the findings.

Reviewer #2 (Comments to the Authors (Required)):

LSA-2020-00693-T

The revised version has improved in many aspects, although there are still some minor aesthetic aspects that could be improved in the figures. I have some minor comments and a concern regarding the new data to report on insulin/PI3K activity that need still clarification.

Comments

(1) I think that there is an inconsistency in the introduction in this new version when introducing the influence of glioma cells on neurons.

In line 107, the authors say: However, whether tumoural glial cells are able to modify insulin signaling remotely in neurons, and consequently alter the number of..."

In line 118, they state "the glial tumor can impact on neighboring neurons.." Neighborhood is not coherent with remotely. I think they should rephrase these sentences or clarify what they mean by a remote effect on 'neighboring' neurons.

(2) "Finally, we propose a novel neuroprotective strategy against GB that extend lifespan and improve life quality". Would this strategy be to introduce a 'transgene' that target expression of human Rheb in the neurons of patients with glioma? I think it would be sensible to eliminate or reformulate this sentence again. It does not help the scientific discoveries to propose "new neuroprotective strategy." When there is not yet base for it or there's really no way to implement it.

(3) In line 221-222, these new results show that PH reporter activity in neurons is strongly reduced upon GB induction (fig 4A-B). The authors should present first the data and only after state that these are new results (see also comment on PH-GFP).

(4) In line 223: "suggesting an up-regulation of insulin signalling activity (fig 4C)". It should say a "normalization or insulin signaling levels".

(5) THOR- MIMIC line. If this line is ThorMI09732, the authors must use the correct name (Thor not THOR, nor thor nor MIMIC-THOR) and they should use the exact name, at least once in the text.

(5) Figure 4. What the authors call PH-GFP report, I presume is tGPH, the reporter based on the GPH gene under the control of the tubulin promoter. If the PH-GFP is something else, please correct the reference and explain what is the PH-GFP used in this study.

This tGPH reporter is used to measure its membrane localization as a proxy for PI3K activation. For this, it is necessary to see the levels of tGPH (e.g. the levels in the membrane and cytoplasm).

The images in Fig. 4 show that overall levels of PH-GFP (tGPH?) are undetectable for reasons that are not known. This is not easy to reconcile with other studies in which it is seen that even under conditions of a strong reduction in insulin signaling (e.g. starvation), the reporter is expressed at detectable levels in the cytoplasm and weak but detectable in the membrane.

The results in Fig. 4 need to be explained. Together with dRheb expression data, it appears that the insulin status in this mutant condition is more complicated than anticipated by the authors. I think the current data do not yet clarify this essential point for the conclusions of the article.

In Fig.4, "the Thor-MIMIC GFP transgene", correct to ThorMIMIC line.

Fig. 5 A) RT-qPCR of repo>UAS-LacZ; elav-lexA>LexAop-CD8GFP (Control) and repo>UAS-

dEGFR^Δ, UAS-dp110CAAX; Elav-lexA>LexAop-CD8GFP (Glioma) and repo>UASdEGFRUAS-dp110CAAX; elav-LexA>LexAop-CD8GFP, LexAOp-Rheb (Glioma+Elav>Rheb) flies shows a down-regulation of Rheb in glioma brains , rescued by ectopical expression of Rheb in neurons (ANOVA, post-hoc Bonferroni).

The graph has only two bars, not three. One control is missing.

Correct all to 'lexAop' or use it consistently. Correct mito-cherry to mito-Cherry.

Fig. 6 Please indicate in the figure legends whether the flies in the survival curve are males or females.

Figure 7 . Please check the title of this figure legend, I presume the authors means: Schematic representation of the effect of GB on the neuronal insulin pathway.

Reviewer #3 (Comments to the Authors (Required)):

Although the authors wrote something in response to each of my questions/concerns in the original review, in many cases they didn't answer the question or address the main issue (e.g. major issues 1, 2, 3 and minor issues 6 and 7).

That said, the manuscript is improved, in that a PtdIns(3,4,5)P3 sensor was used to measure insulin signaling in the neurons, which makes the manuscript significantly stronger. Also, the manuscript does contain a lot of good and interesting data, so that on the whole I support publication of this revised version in Life Science Alliance. It is good and important that the reviews are published alongside the article so that the readership can see what the remaining open issues are in case they become relevant in the future.

Reviewer #1 (Comments to the Authors (Required)):

The issues raised in my review were addressed by the authors. The additional experiments performed and the re-writing of some sections better represent the novelty and importance of the findings.

We would like to thank the reviewer for their help to improve the manuscript.

Reviewer #2 (Comments to the Authors (Required)):**LSA-2020-00693-T**

The revised version has improved in many aspects, although there are still some minor aesthetic aspects that could be improved in the figures. I have some minor comments and a concern regarding the new data to report on insulin/PI3K activity that need still clarification.

Comments

(1) I think that there is an inconsistency in the introduction in this new version when introducing the influence of glioma cells on neurons. In line 107, the authors say: However, whether tumoural glial cells are able to modify insulin signaling remotely in neurons, and consequently alter the number of..."

In line 118, they state "the glial tumor can impact on neighboring neurons.." Neighborhood is not coherent with remotely. I think they should rephrase these sentences or clarify what they mean by a remote effect on 'neighboring' neurons.

We thank the reviewer. He is right, remotely should not be used in this context, given that GB cells impact exclusively in neurons that are nearby.

(2) "Finally, we propose a novel neuroprotective strategy against GB that extend lifespan and improve life quality". Would this strategy be to introduce a

'transgene' that target expression of human Rheb in the neurons of patients with glioma? I think it would be sensible to eliminate or reformulate this sentence again. It does not help the scientific discoveries to propose "new neuroprotective strategy." When there is not yet base for it or there's really no way to implement it.

The reviewer is right, so we have removed it from the introduction.

(3) In line 221-222, these new results show that PH reporter activity in neurons is strongly reduced upon GB induction (fig 4A-B). The authors should present first the data and only after state that these are new results (see also comment on PH-GFP).

We have changed this sentence, removing "new results".

(4) In line 223: "suggesting an up-regulation of insulin signalling activity (fig 4C)". It should say a "normalization or insulin signaling levels".

We have changed this accordingly.

(5) THOR- MIMIC line. If this line is , the authors must use the correct name (Thor not THOR, nor thor nor MIMIC-THOR) and they should use the exact name, at least once in the text.

We have changed this accordingly.

(5) Figure 4. What the authors call PH-GFP report, I presume is tGPH, the reporter based on the GPH gene under the control of the tubulin promoter. If the PH-GFP is something else, please correct the reference and explain what is the PH-GFP used in this study.

This tGPH reporter is used to measure its membrane localization as a proxy for PI3K activation. For this, it is necessary to see the levels of tGPH (e.g. the levels in the membrane and cytoplasm).

The images in Fig. 4 show that overall levels of PH-GFP (tGPH?) are undetectable for reasons that are not known. This is not easy to reconcile with other studies in which it is seen that even under conditions of a strong reduction in insulin signaling (e.g. starvation), the reporter is expressed at detectable levels in the cytoplasm and weak but detectable in the membrane.

The results in Fig. 4 need to be explained. Together with dRheb expression data, it appears that the insulin status in this mutant condition is more complicated than anticipated by the authors. I think the current data do not yet clarify this essential point for the conclusions of the article.

We apologize for the mistake in naming the reporter. Indeed, we used the tGPH the reviewer mentioned, so we have changed the name PH-GFP to tGPH in the text. Regarding the main concern of the referee, he/she is right in pointing out that tGPH fluorescence should be detectable. Actually, we see a weak signal in fig 4B (please find attached the image for the reviewer with enhanced contrast that reveals this fluorescence in most neurons). However, we took the confocal images using the same conditions in order to avoid saturation, which meant that the low tGPH expression in the GB seemed undetectable by eye, but a deeper analysis show that there is signal albeit very weak,

similarly to what happens in starving conditions. We added "(although still detectable)" in line 223.

In Fig.4, "the Thor-MIMIC GFP transgene", correct to ThorMIMIC line.

We have changed Thor-MIMIC to *Thor*^{M109732}

Fig. 5 A) RT-qPCR of repo>UAS-LacZ; elav-lexA>LexAop-CD8GFP (Control) and repo>, UAS-dp110CAAX; Elav-lexAλUAS-dEGFR>LexAop-CD8GFP (Glioma) and repo>UASdEGFRUAS-dp110CAAX; elav-LexA>LexAop-CD8GFP, LexAop-Rheb (Glioma+Elav>Rheb) flies shows a down-regulation of Rheb in glioma brains , rescued by ectopical expression of Rheb in neurons (ANOVA, post-hoc Bonferroni).

The graph has only two bars, not three. One control is missing. Correct all to 'lexAop' or use it consistently. Correct mito-cherry to mito-Cherry.

We have corrected these issues accordingly

Fig. 6 Please indicate in the figure legends whether the flies in the survival curve are males or females.

We have included this information in the figure legend

Figure 7 . Please check the title of this figure legend, I presume the authors means: Schematic representation of the effect of GB on the neuronal insulin pathway.

We have corrected this sentence accordingly

Reviewer #3 (Comments to the Authors (Required)):

Although the authors wrote something in response to each of my questions/concerns in the original review, in many cases they didn't answer the question or address the main issue (e.g. major issues 1, 2, 3 and minor issues 6 and 7).

That said, the manuscript is improved, in that a PtdIns(3,4,5)P3 sensor was used to measure insulin signaling in the neurons, which makes the manuscript significantly stronger. Also, the manuscript does contain a lot of good and interesting data, so that on the whole I support publication of this revised version in Life Science Alliance. It is good and important that the reviews are published alongside the article so that the readership can see what the remaining open issues are in case they become relevant in the future.

We want to thank the reviewer for his/her comments and we agree that the review process should be included along with the manuscript.

January 8, 2021

RE: Life Science Alliance Manuscript #LSA-2020-00693-TRR

Dr. Sergio Casas-Tintó
Instituto Cajal
Developmental Neurobiology
Avda Doctor Arce, 37
Madrid, Madrid 28002
Spain

Dear Dr. Casas-Tintó,

Thank you for submitting your Research Article entitled "Insulin signaling mediates neurodegeneration in glioma". It is a pleasure to let you know that your manuscript is now accepted for publication in Life Science Alliance. Congratulations on this interesting work.

DISTRIBUTION OF MATERIALS:

Again, congratulations on a very nice paper. I hope you found the review process to be constructive and are pleased with how the manuscript was handled editorially. We look forward to future exciting submissions from your lab.

Sincerely,

Shachi Bhatt, Ph.D.

Executive Editor

Life Science Alliance

<https://www.lsjournal.org/>
